# Learning When to Attend: Conditional Memory Access for Long-Context LLMs

Sakshi Choudhary [*1]   Aditya Chattopadhyay [*2]   Luca Zancato [2]   Elvis Nunez [2]   Matthew Trager [2]   Wei Xia [2]   Stefano Soatto [2]

## Abstract

Language models struggle to generalize beyond pretraining context lengths, limiting long-horizon reasoning and retrieval. Continued pretraining on long-context data can help but is expensive due to the quadratic scaling of Attention. We observe that most tokens do not require (Global) Attention over the entire sequence and can rely on local context. Based on this, we propose *L2A* (Learning To Attend), a layer that enables conditional (token-wise) long-range memory access by deciding *when* to invoke global attention. We evaluate *L2A* on Qwen 2.5 and Qwen 3 models, extending their effective context length from 32K to 128K tokens. *L2A* matches the performance of standard long-context training to within 3% while skipping Global Attention for ∼80% of tokens, outperforming prior baselines. We also design custom Triton kernels to efficiently implement this token-wise conditional Attention on GPUs, achieving up to ∼2× improvements in training throughput and time-to-first-token over FlashAttention. Moreover, *L2A* enables post-training pruning of highly sparse Global Attention layers, reducing KV cache memory by up to 50% with negligible performance loss. Our code is released under Apache 2.0 at https://github.com/awslabs/hybrid-model-factory/tree/main/examples/research/L2A.

## 1. Introduction

Consider a 100,000-token document. A sentence discussing "therefore the quarterly revenue is" likely depends mostly on the preceding summary paragraphs, but a phrase like "as mentioned in Section 2" may require retrieving information

*Equal contribution . Work done during an internship at AWS Agentic AI. [1]Department of Electrical and Computer Engineering, Purdue University, USA [2]AWS Agentic AI, USA. Correspondence to: Aditya Chattopadhyay <achatto@amazon.com>.

*Proceedings of the $43^{rd}$ International Conference on Machine Learning*, Seoul, South Korea. PMLR 306, 2026. Copyright 2026 by the author(s).

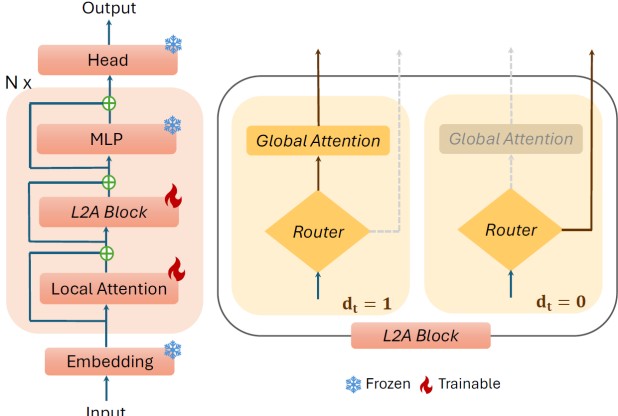

*Figure 1.* **Overview of *L2A*, a sequence modeling layer with token-wise conditional routing.** All tokens are first processed by Local Attention. Next, tokens with routing decision $\mathbf{d}_t = 1$ invoke Global Attention, whereas tokens with $\mathbf{d}_t = 0$ bypass it. This enables efficient long-context modeling by invoking expensive Global Attention only when needed. We refer to the combined {Local Attention + *L2A* Block} as the *L2A* layer.

from thousands of tokens earlier. Standard Attention treats both cases identically, this uniformity is wasteful: most tokens don't need to look far in the past, yet all pay the quadratic cost of Global Attention. This observation suggests the following question: can a model decide, token by token, *when* to access its long-term memory, and not just *what* to retrieve from it?

We are interested in this problem since Large Language Models (LLMs) are being increasingly deployed in long-context scenarios such as multi-turn conversations (Maharana et al., 2024) and debugging long code repositories (Jimenez et al., 2023). However, training long-context LLMs is computationally expensive due to the quadratic complexity of Attention (Liu et al., 2024a), which enables it to perform verbatim lookup over its entire past (Zancato et al., 2024).

Prior work on efficient long-context modeling has focused on a different question: *what* information to retrieve from the past. Methods like Native Sparse Attention (Yuan et al., 2025), Expansion Span (Nunez et al., 2025) and Landmark Attention (Mohtashami & Jaggi, 2023) sparsify the key-value pairs that each query attends to, selecting or compressing a fixed subset of the context. These approaches apply the

same retrieval budget to every token, regardless of whether that token actually needs distant information. The result is a one-size-fits-all sparsity pattern that cannot adapt to the varying complexity of user's queries.

We introduce *L2A*, a sequence modeling layer that adaptively determines, at each token, whether to apply Global Attention (Attention applied to the whole context) or rely solely on local short-context Attention (Figure 1). Specifically, *L2A* first computes Local Attention over the recent past, the resulting output is then passed to a learned routing function that decides whether Global Attention is required. Tokens that trigger Global Attention retrieve information from the full context, while others retain only their local representations. In this way, *L2A* treats Global Attention as a selective long-term memory retrieval mechanism and Local Attention as a short-term memory for capturing immediate contextual dependencies.

*L2A* is trained with a standard next-token prediction objective, along with a regularizer that encourages sparsity in the number of tokens invoking Global Attention. In our experiments, we show that this strategy preserves long-context performance while substantially reducing the computational cost of long-context fine-tuning.

A key distinction between *L2A* and prior sparse or approximate Attention methods lies in where sparsity is applied. Existing approaches compute Attention for every query over a sparsified set of past key–value tokens (Yuan et al., 2025; Chen et al., 2024), whereas *L2A* computes exact Attention over the full context, but only for a sparse subset of query tokens. Hence, as alluded to before, prior work focuses on *what* information to access from the past, our approach focuses on *when* to access it. This enables token-wise conditional Global Attention that adapts to the query complexity, which is often unknown a priori. For instance, in needle-in-a-haystack tasks (Kamradt, 2024), long-term memory access might be needed only at specific token positions, whereas for in-context learning, Global Attention may be triggered frequently to compare and reuse provided examples.

To translate this algorithmic sparsity into real wall-clock speedups, we design custom Triton kernels that yield up to $2\times$ improvement in training throughput and time-to-first-token compared to FlashAttention-2 (FA-2) (Dao, 2023), which applies Global Attention uniformly to all tokens. *L2A* is implemented as a drop-in replacement for standard Attention layers, enabling direct fine-tuning of off-the-shelf LLMs at long context lengths via continued pre-training.

Empirically, we demonstrate that *L2A* extends the context length of Qwen2.5 and Qwen3 models from 32K to 128K tokens. Across long-context benchmarks such as HELMET (Yen et al., 2025), BabiLong (Kuratov et al., 2024), and MRCR (Vodrahalli et al., 2024), *L2A* matches the perfor-

mance of models trained with (exact) Attention over long contexts—a training practice commonly referred to as Continued Long-Context Pre-training (CLP)—within 1.5–3%, while reducing training costs by allowing 75–80% of tokens to skip Global Attention. Moreover, across different model scales (1.5B and 7B), *L2A* consistently outperforms sparse Attention-based training methods (Native Sparse Attention (Yuan et al., 2025), $S^2$-Attn(Chen et al., 2024)) and training-free baselines (Dual Chunk Attention (An et al., 2024), SelfExtend (Jin et al., 2024). Figure 2 presents an overview of these results.

Despite *L2A*'s selectivity and higher efficiency, it requires storing the whole past KV cache, which can still be expensive for very long sequences. To address this, we propose a post-training layer-pruning strategy that retains only a local cache (4K tokens) for layers in which Global Attention is rarely activated (for sparsity $\geq 95\%$). Compared to a similarly-sized Transformer, this reduces KV cache size by up to 50% at 128K context length with minimal performance degradation ($\leq 1\%$).

**Paper Contributions.** (1) We propose *L2A*, an efficient training alternative to extend the context length of LLMs through token-wise conditional access to long-term memory. (2) We design Triton kernels that exploit the sparsity arising from conditional long-term memory access to achieve upto $10\times$ speedups over FA-2 on modern GPUs. (3) Empirical evaluation shows that *L2A* extends the context length of Qwen2.5 and Qwen3 models from 32K to 128K tokens, matching the performance of continued long-context pre-training while being $2\times$ faster to train.

## 2. Preliminaries & Prior Work

We review the Attention mechanism (Vaswani et al., 2017) and its computational cost, followed by a discussion of prior work on sparse Attention variants. We defer further discussion on context length extrapolation, dynamic routing, and external memory-based approaches to Section B.

**Attention Mechanism.** For an input sequence of token embeddings $\mathbf{X} \in \mathbb{R}^{n \times d}$, where $n$ denotes the sequence length and $d$ is the hidden dimension, the model computes query, key, and value representations as:

$$\mathbf{Q} = \mathbf{X}\mathbf{W^Q}, \quad \mathbf{K} = \mathbf{X}\mathbf{W^K}, \quad \mathbf{V} = \mathbf{X}\mathbf{W^V} \quad (1)$$

Here, $\mathbf{W^Q}, \mathbf{W^K}, \mathbf{W^V} \in \mathbb{R}^{d \times d}$ are learnable projection matrices. At each timestep, the Attention mechanism computes a weighted average over all previously observed values, where the weights are determined by an exponentiated and normalized inner product between the current query and past keys. This can be written compactly for all tokens as:

$$\text{Attn}(\mathbf{Q}, \mathbf{K}, \mathbf{V}) = \text{softmax}\left(\frac{\mathbf{Q}\mathbf{K}^\top}{\sqrt{d}} + M\right)\mathbf{V}, \quad \text{(Attn)}$$

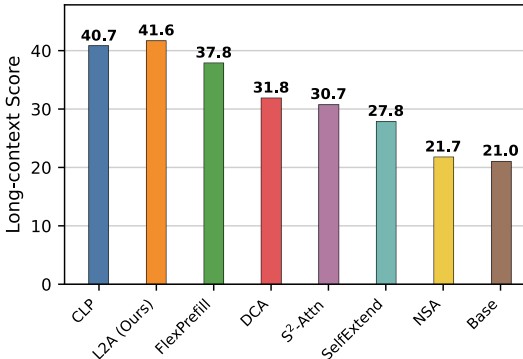
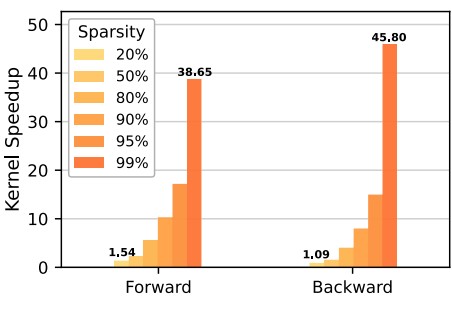

*Figure 2.* **Long-context performance and efficiency of *L2A* at 128K context length.** *Left*: Performance comparison of *L2A* against CLP for base model and other baselines on Qwen 2.5 7B, where *L2A* outperforms existing approaches. *Right*: Kernel-level speedups achieved by *L2A* relative to FlashAttention-2, with $\sim 10\times$ and $\sim 8\times$ speedups for the forward and backward passes at upto 90% sparsity in practice.

where, $M_{ij} = 0$ if $j \leq i$ and $-\infty$ otherwise, enforcing causal Attention for autoregressive language modeling.

The pairwise similarity matrix $\mathbf{QK}^\top$ scales quadratically with sequence length, making it a major computational bottleneck for long-context training. We refer to (Attn) as Global Attention to distinguish it from Local Attention variants, such as Sliding Window Attention, which restrict Attention to a fixed window of recent keys and values. Mathematically, this corresponds to modifying the mask in (Attn) by setting $M_{ij} = 0$ if $0 \leq i - j \leq w$ and $-\infty$ otherwise, where $w$ is the window size, restricting each token to attend to itself and the $w$ preceding tokens.

**Sparse (Approximate) Attention Variants.** Various sparse and approximate Attention variants have been proposed to reduce the computational cost of Attention and broadly fall into trainable and training-free approaches.

Trainable sparse Attention methods reduce training cost by compressing keys (Munkhdalai et al., 2024), selecting landmarks (Mohtashami & Jaggi, 2023; Nunez et al., 2025), or imposing fixed sparse patterns (Chen et al., 2024). Approaches such as S²-Attn (Chen et al., 2024) and Landmark Attention (Mohtashami & Jaggi, 2023) improve efficiency through sparse or summarized access to past context, but often fail to capture exact long-range dependencies, leading to degraded performance on challenging tasks (Gao et al., 2025; Lu et al., 2024). SE-Attention (Nunez et al., 2025) improves efficiency of Transformer architectures by replacing Global Attention with a Local Attention that compresses and retrieves blocks of tokens from the past. Similarly, Native Sparse Attention (NSA) (Yuan et al., 2025) combines compressed, selected, and local context within a sliding window. These methods assume a fixed budget per query, attending to a predetermined set of key blocks regardless of contextual needs. As a result, sparsity is fixed and does not adapt to input or task complexity, which can lead to sub-optimal long-context performance, since the amount of relevant past information is difficult to determine a priori during training

(see Section 1 for some examples). Importantly, *L2A* is complementary to these approaches and could potentially be combined with them in a two-stage mechanism, where *L2A* first identifies query tokens requiring long-range context, followed by sparse key/value selection over the relevant past context. Such a combination could jointly reduce both the number of active queries and the attention span per query.

In contrast, training-free sparse Attention methods primarily target inference efficiency by reducing Attention cost during prefill and decoding, for example, through FlexPrefill and MInference for prefill acceleration and H2O and TokenSkip for faster decoding (Lai et al., 2025; Jiang et al., 2024a; Zhang et al., 2023; Xia et al., 2025). These approaches focus on inference-time optimization without providing training-time efficiency benefits.

In this work, we view Global Attention as a long-term memory retrieval mechanism and train models to learn *when* to access it in a token-wise, context-dependent manner. A routing function determines whether Global Attention is required beyond local context, enabling adaptive retrieval based on input complexity.

## 3. L2A: Learning When to Attend

Motivated by recency bias in LLMs (Peysakhovich & Lerer, 2023), *L2A* primarily relies on local context and invokes Global Attention only when necessary.

### 3.1. The *L2A* Layer

As illustrated in Figure 1, *L2A* consists of three modules: Local Attention, a Router, and Global Attention. To enable seamless integration with existing LLM architectures, Local Attention is implemented using Sliding Window Attention (SWA), while Global Attention is realized using (Attn) applied over the full context. This design allows both modules to be initialized from the Attention layer parameters of an

existing LLM, which we refer to as the Base LLM. The Router is implemented as a linear projection followed by a sigmoid activation, with weights initialized to zero to trigger Global Attention for all tokens at initialization.

**Base LLM → *L2A* LLM.** We replace all Attention layers in Base LLM with *L2A* layers using the procedure described above and leave the Feed-Forward Network (FFN) layers unchanged. We refer to the resulting model as the *L2A* LLM, which is subsequently trained to support longer context lengths. Incorporating both Local and Global Attention introduces only a modest increase in parameter count (about 10% for the Qwen2.5 models) compared to the Base LLM, as the majority of parameters reside in the FFN layers.

**Forward Pass for *L2A* Layer**: Given an input sequence $(\mathbf{x}_1, \ldots, \mathbf{x}_n)$, each token $\mathbf{x}_t$ first attends to a local context of size $k \ll n$ using SWA:

$$\mathbf{s}_t = \text{LocalAttn}(\mathbf{x}_t, \mathbf{x}_{t-1}, \ldots, \mathbf{x}_{t-k}), \quad t = 1, \ldots, n. \quad (2)$$

This local representation, which we denote $\mathbf{s}_t$, is then passed to the Router module, which computes a score $\hat{\mathbf{d}}_t$ in $[0, 1]$ that is then thresholded to obtain the Router's decision $\mathbf{d}_t$:

$$\hat{\mathbf{d}}_t = \sigma(\mathbf{W}\mathbf{s}_t), \quad \mathbf{d}_t = \begin{cases} 1, & \text{if } \hat{\mathbf{d}}_t \geq 0.5 \\ 0, & \text{otherwise} \end{cases} \quad (3)$$

Here, $\mathbf{W} \in \mathbb{R}^{1 \times d}$ denotes a learnable linear projection for the Router and $\sigma(.)$ refers to the Sigmoid activation function. If $\mathbf{d}_t = 1$, the token is processed also via Global Attention, otherwise, it retains its local representation $\mathbf{s}_t$.

$$\mathbf{a}_t = \begin{cases} \text{GlobalAttn}(\mathbf{s}_t, \mathbf{s}_{t-1}, \ldots, \mathbf{s}_1), & \text{if } \mathbf{d}_t = 1 \\ 0, & \text{otherwise} \end{cases} \quad (4)$$

The final output is thus given as[1]:

$$\mathbf{o}_t = \mathbf{s}_t + (\mathbf{a}_t \cdot \mathbf{d}_t) \quad (5)$$

Having described the *L2A* layer, we note that implementing such a conditional computation scheme in LLM training introduces a few technical challenges:

1. GPUs are optimized for dense matrix multiplications, enabling highly efficient kernels for Global Attention (Dao, 2023). In contrast, sparse computations arising from conditional execution are harder to accelerate due to irregular memory access and poor memory coalescing (Li et al., 2025; Gale et al., 2020; Gray et al., 2017).

---

[1]As is standard practice, we employ LayerNorm operations after each Local and Global Attention modules; we omit this detail from the exposition for brevity.

2. Training LLMs with input-conditional computation graphs can be unstable due to discontinuous routing decisions, leading to *Router collapse*, as observed in Mixture-of-Experts models (Wang et al., 2024b).

We address these challenges by: (i) implementing custom Triton kernels that build upon the tiled attention computation of FA-2 in a sparsity- and hardware-aware manner (see Section 3.2); and (ii) proposing a training algorithm that ensures stability and prevents convergence to solutions that rely only on local context (see Section 3.3). We elaborate on each of these two components in the following subsections.

### 3.2. *L2A* Kernel Design

To translate algorithmic sparsity into wall-clock speedups, we design a custom Triton (Tillet et al., 2019) kernel to realize conditional Global Attention by building upon the tiled Attention computations introduced in FA-2 (Dao, 2023). FA-2 partitions the $\mathbf{Q}$, $\mathbf{K}$, and $\mathbf{V}$ tensors into tiles, which are then streamed from high-bandwidth memory (HBM) into fast on-chip SRAM. For each $\mathbf{Q}$ tile, the kernel iterates over $\mathbf{K}$–$\mathbf{V}$ tiles, applies causal masking, and accumulates attention outputs using an online log-sum-exp scheme, storing the statistics in $L_c$ to ensure numerical stability. By avoiding materialization of the full Attention matrix, FA-2 achieves high arithmetic intensity and GPU utilization.

Building on this design, our kernel supports token-level conditional sparsity by selectively computing Attention only for queries that trigger Global Attention. Active queries (that is, queries with corresponding $\mathbf{d}_t = 1$ in Equation (3)) are first consolidated into a compact, contiguous buffer $\mathbf{Q}_c$, enabling efficient FA-2-style tiling, while their original sequence positions are recorded in an index mapping $\mathbf{Q}_{idx}$ to preserve causality. Attention is computed over tiles of $\mathbf{Q}_c$, $\mathbf{K}$, and $\mathbf{V}$, with causal masking enforced using the true query indices in $\mathbf{Q}_{idx}$ rather than the compacted query order. This allows the kernel to skip key–value tiles outside a query's causal range, reducing unnecessary computation while maintaining correctness. As in FA-2, Attention scores are accumulated using the same log-sum-exp scheme. After processing all relevant key–value tiles, $\mathbf{O}_c$ and $L_c$ are written back to HBM, and the full output tensor $\mathbf{O}$ is reconstructed from $\mathbf{O}_c$ using $\mathbf{Q}_{idx}$.

This design preserves the memory efficiency and numerical stability of FA-2, while enabling conditional sparsity that translates directly into wall-clock speedups. The forward pass is detailed in Algorithm 1 in the Appendix. The backward pass follows a similar design, performing gradient computation only for active queries identified by $\mathbf{Q}_{idx}$. The saved statistics $L_c$ and index mapping $\mathbf{Q}_{idx}$ are reused to ensure numerical stability and efficient gradient accumulation without revisiting irrelevant key–value tiles.

### 3.3. *L2A* Training

**Training Objective.** For training *L2A*, we augment the training loss with a sparsity regularizer that penalizes frequent invocation of Global Attention:

$$\mathcal{L} = \mathcal{L}_{\text{NTP}} + \frac{\lambda}{nL} \sum_{l=1}^{L} \sum_{t=1}^{n} \hat{\mathbf{d}}_{l,t}^2, \qquad (6)$$

where $\mathcal{L}_{\text{NTP}}$ denotes the Next Token Prediction (NTP) loss, $\hat{\mathbf{d}}_{l,t}^2$ denotes the Router's output at layer $l$, position $t$, $L$ is the number of *L2A* layers, and $n$ is the sequence length.

As discussed in Section 3.1, the router decision $\mathbf{d}_t$ described in Equation (3) is discrete and non-differentiable, with zero gradients almost everywhere. Hence, we employ the straight-through estimator (STE) (Bengio et al., 2013) to enable backpropagation and obtain a learning signal for the Router. This implies that, during the forward pass, the discrete routing decision $\mathbf{d}_t$ is used; however, in the backward pass, gradients are computed using the sigmoid-based continuous relaxation $\hat{\mathbf{d}}_t$.

**Training Instabilities.** Naively optimizing Equation (6) with stochastic gradient descent results in *Router collapse*, where Global Attention is never invoked for any token. This behavior can be understood by analyzing the gradient of the loss in Equation (6) with respect to the Router score $\hat{\mathbf{d}}_t$:

$$\frac{\partial \mathcal{L}}{\partial \hat{\mathbf{d}}_t} = \left( \frac{\partial \mathcal{L}_{\text{NTP}}}{\partial \mathbf{o}_t} \right)^T \mathbf{a}_t + \frac{2\lambda}{nL} \hat{\mathbf{d}}_t. \qquad (7)$$

Recall from Section 3.2 that the Global Attention outputs $\mathbf{a}_t$ (Equation (4)) are computed only for query tokens that invoke it. For all other tokens, referred to as *masked* tokens, $\mathbf{a}_t = 0$. Consequently, $\hat{\mathbf{d}}_t$ for these tokens receives no gradient signal from the NTP loss. As training progresses, the regularization term then dominates and drives $\mathbf{d}_t \to 0$ for all tokens, causing the Router to completely deactivate Global Attention. A detailed derivation for Equation (7) and subsequent discussion is provided in Section A.

To counteract this missing learning signal for the masked tokens, we invoke Global Attention for all tokens (regardless of the Router's decision) with a small probability.[3] This strategy preserves the forward behavior for the masked tokens since the Global Attention output, $\mathbf{a}_t$, is always multiplied by $\mathbf{d}_t = 0$ (see Equation (5)), while ensuring that non-zero gradients from the NTP loss reach the Router during backpropagation, even for the masked tokens. Since Global Attention is invoked sparingly, *L2A* largely retains the training throughput benefits of conditional computation.

### 3.4. *L2A* Inference

Autoregressive inference consists of two stages: prefill and decode. During prefill, Local Attention uses standard SWA kernels from FA-2, while Global Attention employs our sparse kernel to selectively compute attention for active queries. During decode, both modules use standard FA-2 decode kernels, with Global Attention skipped entirely for tokens where $\mathbf{d}_t = 0$.

**The KV Cache Bottleneck During Inference.** At long context lengths, storing the KV cache becomes the primary memory bottleneck (Saxena et al., 2024; Liu et al., 2024c; Pope et al., 2023). In *L2A*, even when Global Attention is skipped for a given token, its key and value representations must still be stored, as they may be required by future tokens that invoke Global Attention. Consequently, without further optimization, *L2A* and the Base LLM incur similar KV cache costs during decode. The additional KV storage from Local Attention is negligible (less than 3% of the KV footprint for a 4K local window at 128K context). However, as we describe next, we can leverage *L2A*'s sparse Global Attention usage to further reduce KV cache memory at inference time.

**Sparsity-based Layer Pruning.** To reduce this overhead, we introduce a post-training layer pruning strategy that leverages the Global Attention sparsity measured at the end of training. For layers exhibiting high sparsity (measured on held-out data), we prune the Global Attention module entirely while retaining Local Attention. This approach is particularly effective because Local Attention stores KV states only within a small sliding window, whereas Global Attention requires storage that scales with the full context length—making it the dominant contributor to KV cache memory. Empirically, we find that nearly 50% of Global Attention layers meet our sparsity criterion for pruning, enabling up to a 50% reduction in KV cache memory relative to the Base LLM with less than 1% degradation in downstream performance. These memory savings translate directly into practical benefits as lower KV cache requirements enable larger-batch inference and higher throughput.

## 4. Experiments

In this section, we demonstrate the efficacy of *L2A* for extending LLM context length. We evaluate long-context performance across multiple benchmarks and compare against baselines (Section 4.1), report runtime speedups achieved by our custom Triton kernels over FA-2 (Section 4.2), and present ablations and sparsity analysis (Section 4.4 and 4.5).

**Base LLMs and Evaluation Tasks.** We consider the Qwen family of models as our Base LLMs for context extension, specifically Qwen2.5 7B (Qwen et al., 2025) and Qwen3 8B. For ablations and sparsity analysis, we additionally use

---

[2]we drop $l$ from subsequent notation for brevity.

[3]Specifically, in each training iteration, we sample a Bernoulli (with $p = 0.1$ in our experiments). If heads, we invoke Global Attention for all tokens regardless of $\mathbf{d}_t$. If tails, we follow Equation (4).

Qwen2.5 1.5B to enable cost-efficient experimentation. All models have a pre-trained context length of 32K. Long-context performance is evaluated on HELMET (Yen et al., 2025), BABILong (Kuratov et al., 2024), and MRCR (Vo-drahalli et al., 2024), which cover diverse real-world and synthetic tasks (details in Section F.2).

**Baselines.** We compare *L2A* with training-free context extension methods (SelfExtend (Jin et al., 2024), DCA (An et al., 2024)), training-based sparse attention methods ($S^2$-Attn (Chen et al., 2024), NSA (Yuan et al., 2025)), and an inference-time sparsity method (FlexPrefill (Lai et al., 2025)). Official implementations are used when available. For NSA, we evaluate a widely used open-source version (Li, 2025) (see Section E). We report CLP-trained Base models as upper bound and Base LLM evaluated at long context without further training as a lower-bound reference.

**Implementation Details.** For all training-based experiments, we initialize from the Base LLM and train on a mixture of long documents from publicly available datasets. *L2A* is initialized as described in Section 3.1, and unless stated otherwise, all *L2A* layers use Sliding Window Attention (SWA) with a 4K window. We apply the same initialization strategy to training baselines, initializing sparse attention layers from the Base LLM's Attention parameters. Following Qwen (Yang et al., 2025), we increase the Rotary Position Embedding base frequency from 1M to 5M and train at 128K context length (a four-fold increase over the Base LLM). During training, we freeze FFN layers and update only token-mixing layers (Attention, NSA, *L2A*, and $S^2$-Attn) along with LayerNorm parameters, which we find performs better than full-parameter training (Section E.1.4). FlexPrefill is applied to the CLP model, yielding faster Time-To-First-Token (TTFT) than CLP alone due to prefill sparsity. Additional details are provided in Appendix F.

### 4.1. *L2A* Across Long & Short Context Tasks

We evaluate *L2A* on Qwen2.5 1.5B and 7B as Base LLMs. Across long-context benchmarks and context lengths from 8K to 128K, *L2A* achieves performance within 3% of CLP for both models, demonstrating robust scaling with model size. *L2A* also consistently outperforms all considered training-based and training-free baselines (Figure 4). While most methods match Base LLM performance within the pretrained 32K context, their performance degrades, often substantially, beyond this length relative to CLP. In contrast, *L2A* maintains strong performance beyond the pretrained context, highlighting the benefit of conditionally accessed long-term memory over fixed sparsity levels. While NSA matches full-attention performance when trained from scratch (Yuan et al., 2025), prior work shows significant degradation under continued pretraining, even within the native context length (Zhao et al., 2025), which aligns with

our results in Figure 4.

We further report task-wise performance at 128K context, including Synthetic Recall, Retrieval-Augmented Generation, and In-Context Learning (Figure 3). *L2A* consistently outperforms all baselines and remains competitive with CLP (detailed results in Tables 8 and 9). We additionally evaluate *L2A* on Qwen3 8B and observe similar trends, with *L2A* outperforming all baselines while remaining within 3% of CLP (see Table 10). These results demonstrate the utility of *L2A* across model families. Overall, *L2A* is the only method that achieves near-zero performance gap relative to CLP on long-context tasks while also achieving higher training throughput due to sparse use of Global Attention.

To assess the impact of long-context training on short-context performance, we also evaluate short-context benchmarks with sequence lengths below 2K tokens. As shown in Table 1, *L2A* closely matches CLP across both model scales, with only small differences consistent with seed-level variability. This indicates that training *L2A* at 128K context length does not degrade short-context performance.

### 4.2. Runtime Speedups from *L2A* Kernels

We begin by evaluating the throughput of the standalone *L2A* kernel on NVIDIA H200 GPUs by benchmarking the forward and backward passes of our custom Triton kernel against FlashAttention-2 (FA-2). As shown in Figure 9 (Section D.1), our kernel achieves 1.6-35× and 1.08-45× speedups for the forward and the backward pass across varying sparsity levels, where sparsity denotes the fraction of queries for which Global Attention is skipped.

**Training Speedups.** These kernel-level gains translate into higher training throughput for *L2A* compared to CLP by conditionally invoking Global Attention only when required, thereby reducing the cost of the most expensive operation at long context lengths. As a result, across different Base LLMs, *L2A* achieves nearly 2× higher training throughput at 128K context length. Detailed throughput comparisons are reported in Table 2. Compared to other sparse Attention baselines, *L2A* provides the best tradeoff between efficiency and long-context performance. We demonstrate this on Qwen2.5-7B, where $S^2$-Attn achieves higher throughput (∼1725 tokens/GPU/s versus ∼1384 tokens/GPU/s for *L2A*) but exhibits a 10-point performance gap from CLP (Figure 2). NSA achieves substantially lower throughput (∼565 tokens/GPU/s) and a larger 20.5-point gap from CLP. In contrast, *L2A* remains close to CLP performance while improving training throughput, making it Pareto-optimal among the evaluated baselines.

**Inference Speedups.** As described in Section 3.4, the *L2A* kernel is applied during the prefill phase of inference and can therefore exploit sparsity. Similar to training, *L2A* achieves

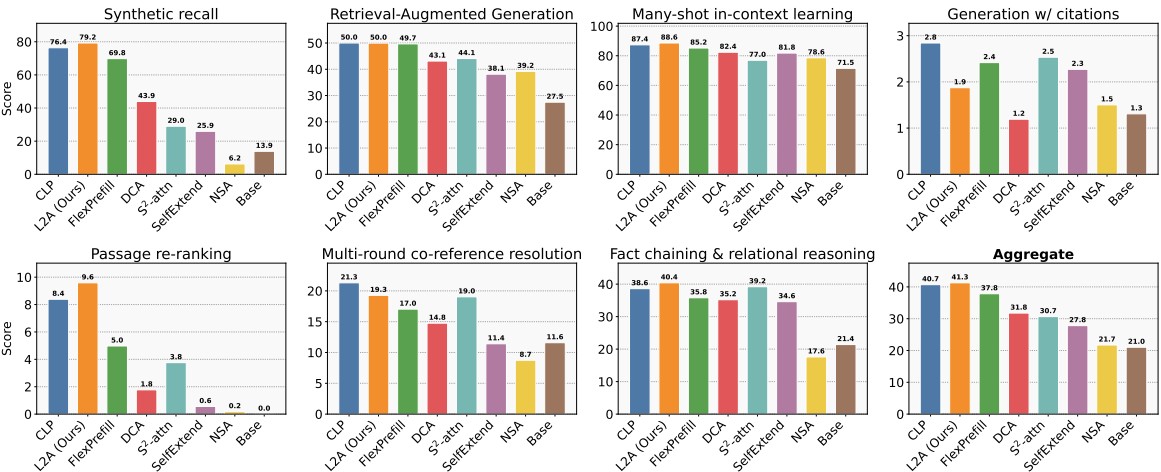

*Figure 3.* Task-wise performance comparison of *L2A* and baseline methods on Qwen 2.5 7B model. *L2A* remains close to CLP performance while outperforming baselines on the majority of tasks.

*Table 1.* Zero-shot performance on standard short-context benchmarks for Qwen 2.5 models at different scales. *L2A* closely matches CLP, indicating that training at 128K context length does not degrade short-context performance

| Scale | Method | BoolQ acc ↑ | CommSenseQA acc ↑ | PIQA acc_n ↑ | Winogrande acc ↑ | ARC-E acc_n ↑ | ARC-C acc_n ↑ | MMLU acc ↑ | SWDE contains ↑ | Avg |
|-------|--------|------|-------------|------|------------|-------|-------|------|--------|-----|
| 1.5B | CLP | **73.36** | **74.45** | 75.95 | **64.72** | **88.72** | **73.81** | **60.46** | 86.50 | **74.74** |
|      | *L2A* | 71.44 | 73.46 | **76.01** | 64.09 | 88.59 | 73.12 | 60.03 | **87.31** | 74.26 |
| 7B | CLP | **82.57** | **84.52** | 79.43 | **76.01** | 96.09 | **88.65** | **73.39** | 91.00 | **83.95** |
|    | *L2A* | 80.15 | 83.70 | **80.14** | 75.69 | **96.21** | 88.14 | 72.76 | **92.17** | 83.62 |

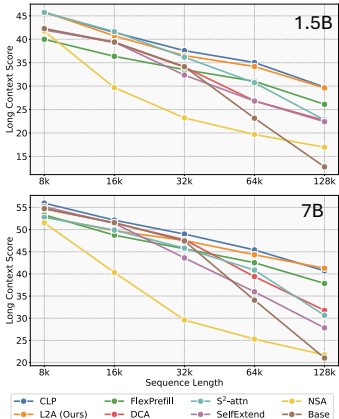

*Figure 4.* Aggregate long-context performance of Qwen models at 1.5B and 7B scales. *L2A* achieves comparable performance to CLP across several context lengths, while significantly outperforming prior baselines.

nearly 2× speedups in time-to-first-token (TTFT) compared to CLP. In Figure 5, we report average TTFT at 64K and 128K context lengths, averaged across tasks from our long-context benchmark suites. The results show that L2A is nearly Pareto-optimal, achieving aggregate long-context performance on par with CLP while approaching the TTFT of FlexPrefill, the fastest baseline in terms of average TTFT.

*Table 2.* *L2A* achieves significantly higher training throughput than the Base LLM across model sizes. We report throughput in tokens per GPU per second, along with the corresponding speedup ratios, for three model variants. All models are trained on NVIDIA H200 GPUs, using batch sizes of 6M tokens for Qwen2.5 7B and Qwen3 8B, and 4M tokens for Qwen2.5 1.5B.

| Model | Method | Toks/GPU/s | Speedup Ratio |
|-------|--------|-----------|---------------|
| Qwen2.5-1.5B | *L2A* (Ours) | 3058.70 | 2.02 |
|              | CLP | 1517.74 | 1.00 |
| Qwen2.5-7B | *L2A* (Ours) | 1383.85 | 2.15 |
|            | CLP | 642.67 | 1.00 |
| Qwen3-8B | *L2A* (Ours) | 410.42 | 1.82 |
|          | CLP | 226.04 | 1.00 |

In the Appendix (Figure 13 and Figure 14), we further analyze how this trend evolves with increasing context length. Notably, *L2A* becomes faster for both training and inference prefill as context length increases, since the cost of Global Attention grows with sequence length.

### 4.3. Leveraging Sparsity to Reduce *L2A*'s KV Cache

In long-context deployment, the KV cache becomes a severe memory bottleneck. While *L2A* uses conditional Global Attention to reduce computation, it still requires the full KV cache (as discussed in Section 3.4). Interestingly, we find

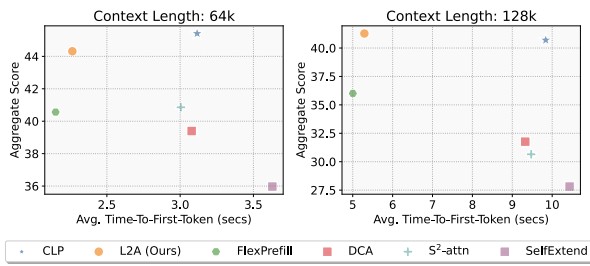

*Figure 5.* Time-to-first-token (TTFT) for Qwen 2.5 7B averaged over different tasks from our long-context benchmark suites. *L2A* achieves similar aggregate long-context performance as CLP, while being ∼ 2× faster.

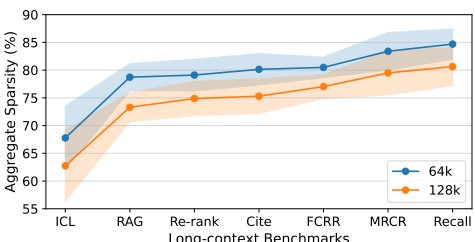

*Figure 6.* Average sparsity for each task at 64k and 128k context lengths, aggregated across Qwen2.5-1.5B, Qwen2.5-7B, and Qwen3-8B models. Tasks are ordered by increasing sparsity. Refer to Section F.2 for task details.

that many layers—nearly 50% in the Qwen 2.5 1.5B *L2A* model- rely almost exclusively on Local Attention (≥95% of the time; see Figure 7). Based on this observation, we conduct the following experiment on the trained Qwen 2.5 1.5B *L2A* model. We compute the average Router sparsity for each layer on a small subset of the training data and remove the Global Attention module from any layer with sparsity exceeding 95%. As shown in Figure 18, this results in nearly 50% KV cache memory savings (15 out of 28 Global Attention modules are dropped) with almost no loss in long-context performance. We further evaluate this pruning strategy on the Qwen2.5-7B *L2A* model using the same 95% sparsity threshold. As shown in Table 7, nearly 40% of the Global Attention layers can be pruned while maintaining performance close to both CLP and the unpruned *L2A* model across context lengths up to 128K. These results suggest that sparsity-driven layer pruning generalizes across model scales. This translates to significant decoding throughput improvements over the CLP model (which has the same KV cache size as *L2A* without layer dropping) since we can now approximately double the batch size.

### 4.4. *L2A*'s Conditional Global Attention Access Patterns

We now analyze the sparsity levels induced by different long-context tasks using our trained *L2A*-based models. In our formulation, sparsity reflects how often Global Atten-

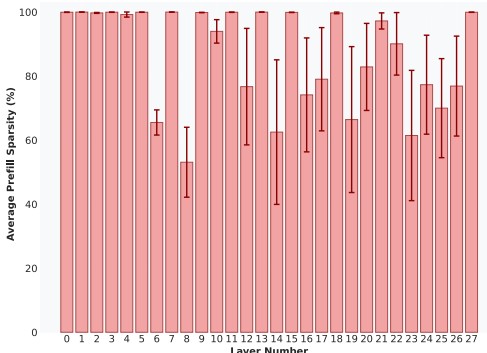

*Figure 7.* Sparsity levels across different layers of Qwen2.5-1.5B *L2A* model. Nearly 50% layers rely almost exclusively (≥ 95%) on Local Attention.

*Table 3.* Average long-context performance of CLP, *L2A*, and *L2A*-shared (shared QKVO projection layers) with Qwen2.5-1.5B. Both *L2A* variants remain close to CLP across context lengths, indicating that parameter sharing does not degrade performance.

| Method | 8k | 16k | 32k | 64k | 128k |
|---|---|---|---|---|---|
| CLP | 45.7 | 41.4 | 37.6 | 35.0 | 29.8 |
| L2A | 45.9 | 40.7 | 36.5 | 34.2 | 29.6 |
| L2A-shared | 43.8 | 40.2 | 36.0 | 32.7 | 27.6 |

tion is triggered for a query token, that is the local context is insufficient for prediction. The task-wise sparsity ordering shown in Figure 6 arises from how this insufficiency is distributed across queries. We additionally observe that, for a fixed task, sparsity generally decreases as the available context length increases. Tasks such as Recall and MRCR (Vodrahalli et al., 2024) concentrate long-range dependencies at a small number of specific positions, making local context sufficient for most tokens and resulting in high sparsity. Retrieval and reasoning tasks like RAG and Passage Re-ranking require global access at multiple tokens for relevance comparison and aggregation, leading to intermediate sparsity. In contrast, In-Context Learning requires comparing patterns across multiple demonstrations, so local context is frequently insufficient at many query positions, resulting in lower sparsity.

### 4.5. Ablation Study

In this subsection, we ablate *L2A*'s design choices.

**Can QKVO projections be shared between Local and Global Attention?** We evaluate a variant of our method with shared QKVO projection layers using Qwen2.5-1.5B as the base LLM. We compare the HELMET (Yen et al., 2025) performance of CLP, *L2A*, and *L2A*-shared (*L2A* with shared QKVO projections). As shown in Table 3, *L2A*-shared remains competitive with *L2A* and close to CLP across context lengths. The small performance gap is consistent with the

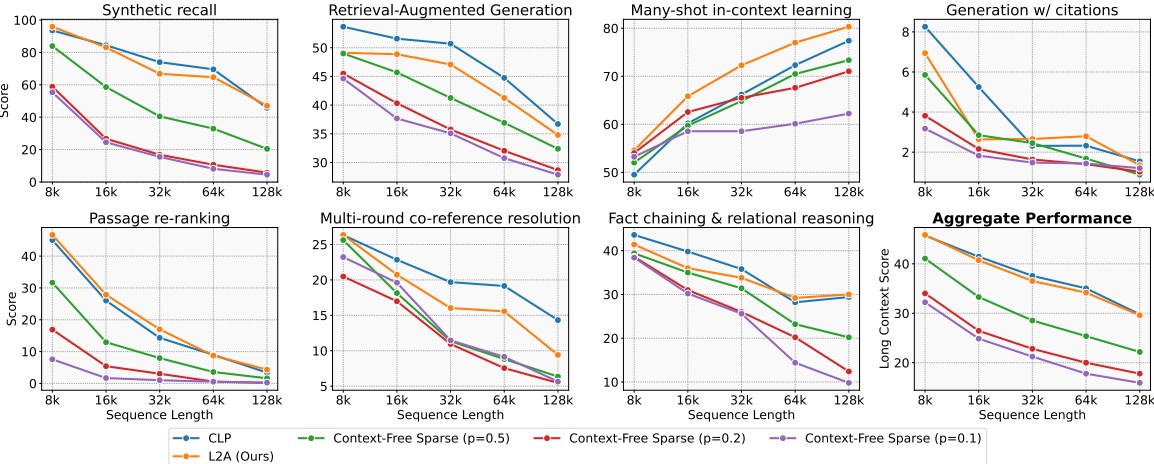

*Figure 8.* Context-free Sparse baselines that independently trigger Global Attention at each token with Bernoulli probability $p \in \{0.1, 0.2, 0.5\}$, corresponding to 50-90% sparsity levels. *L2A* outperforms these by a large margin, providing crucial evidence that the Router leverages local context to decide upon the invocation of Global Attention.

variance reported for this benchmark, indicating that parameter sharing does not significantly degrade performance.[4] However, *L2A*-shared achieves up to 10% lower sparsity than *L2A*. We attribute this to shared projections jointly supporting both Local and Global Attention, limiting their ability to specialize for selective routing.

**Does the Router in *L2A* use context to decide when to invoke Global Attention?** We evaluate this by comparing against a context-free routing baseline that independently triggers Global Attention at each token with Bernoulli probability $p$, which we refer to as *Context-free Sparse*. We consider $p \in \{0.1, 0.2, 0.5\}$, corresponding to 50-90% sparsity levels. Across all settings, *L2A* outperforms this baseline by a large margin, providing strong evidence that the Router leverages local context to decide when Global Attention is required. For example, on Qwen2.5 1.5B, *L2A* achieves approximately 80% average sparsity across all long context tasks while improving performance by nearly 30% over the strongest Context-free Sparse baseline (with $p = 0.5$, 50% sparsity). Results are reported in Figure 8.

**What is the effect of window size in the Local Attention Module in *L2A*?** We vary the SWA window size from 0 to 4K. As shown in Figure 15, performance remains comparable across all configurations. Interestingly, the Router module compensates for smaller SWA windows by reducing sparsity in Global Attention invocations. In other words, as the short-term local context shrinks, *L2A* increasingly relies on long-term global memory. In the extreme case, the SWA-0 configuration (which essentially has only the Router and Global Attention Modules) exhibits 0% sparsity during

---

[4]See discussion in HELMET GitHub issue: `https://github.com/princeton-nlp/HELMET/issues/7#issuecomment-2435378761`.

training (as shown in Figure 16). While accuracy remains comparable, both training and inference efficiency degrade as reliance on Global Attention increases. In particular, the SWA-0 variant incurs approximately a 2× higher time to first token (TTFT) compared to SWA-4K (Figure 17).

Beyond the above ablations we also consider the effect of $\lambda$ (in (6)) on *L2A* performance and efficiency. As expected, increasing $\lambda$ increases sparsity at the cost of long-context performance. We discuss this in Section E.1). However, it is difficult to know a priori what level of $\lambda$ to set for a desired performance and sparsity-level. Training with different value quickly adds to training cost at scale. Instead, we show that one can modulate the sparsity level at test-time by modulating the threshold of the sigmoid in Equation (3) (which is set to $0.5$ during training) for different downstream tasks. Refer to Section E.1.3 for more details.

## 5. Discussion

This work demonstrates that conditional long-term memory access provides a practical path toward efficient long-context modeling. A natural extension of *L2A* is to alternatively consider State-Space Models (SSMs) (Gu & Dao, 2024; Yang et al.; Peng et al., 2025) for Local Attention module. SSMs offer linear-time compute and memory complexity and maintain a "fading" summary of the distant past, enabling them to model both global and local dependencies. However, replacing Attention layers with SSMs in a pretrained LLM typically requires an additional distillation step (Wang et al., 2024a), which can increase training cost and further complicate the *L2A* design. We leave this exploration for future work.

## Impact Statement

This paper presents work whose goal is to advance the field of Machine Learning. There are many potential societal consequences of our work, none which we feel must be specifically highlighted here.

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

## A. Gradient Analysis of the Routing Mechanism

We repeat the equations presented in Section 3.3 here before diving into understanding the routing decision behavior during training. Formally, given an input sequence $(\mathbf{x}_1, \ldots, \mathbf{x}_t)$:

$$\mathbf{s}_t = \text{LocalAttn}(\mathbf{x}_t, \mathbf{x}_{t-1}, \ldots, \mathbf{x}_{t-k}) \ \forall \ t = 1, \ldots, n \tag{8}$$

$$\hat{\mathbf{d}}_t = \sigma(\mathbf{W}\mathbf{s}_t), \quad \mathbf{d}_t = \begin{cases} 1, & \text{if } \hat{\mathbf{d}}_t \geq 0.5, \\ 0, & \text{otherwise.} \end{cases} \tag{9}$$

$$\mathbf{a}_t = \text{GlobalAttn}(\mathbf{s}_t, \mathbf{s}_{t-1}, \ldots, \mathbf{s}_1), \tag{10}$$

The contribution of global attention to the residual stream can be written as

$$\mathbf{o}_t = \mathbf{d}_t . \mathbf{a}_t + \mathbf{s}_t, \tag{11}$$

where this term is added to the residual connection following standard Transformer conventions. Thus, when $\mathbf{d}_t = 0$, the token retains its local representation $\mathbf{s}_t$.

The overall training objective is defined as:

$$\mathcal{L} = \mathcal{L}_{\text{NTP}} + \frac{\lambda}{nL} \sum_{l=1}^{L} \sum_{t=1}^{n} \hat{\mathbf{d}}_{l,t}^2. \tag{12}$$

For clarity, gradients for a single layer are derived, and the layer index $l$ is omitted in what follows. The gradient $\frac{\partial \mathcal{L}}{\partial \mathbf{W}}$ determines how the routing function learns to activate global attention during training. Aggregating contributions across all tokens, we have

$$\frac{\partial \mathcal{L}}{\partial \mathbf{W}} = \sum_{t=1}^{n} \frac{\partial \mathcal{L}}{\partial \hat{\mathbf{d}}_t} \frac{\partial \hat{\mathbf{d}}_t}{\partial \mathbf{W}} \tag{13}$$

$$\frac{\partial \hat{\mathbf{d}}_t}{\partial \mathbf{W}} = \hat{\mathbf{d}}_t (1 - \hat{\mathbf{d}}_t) \mathbf{s}_t^\top \tag{14}$$

The gradient of the loss with respect to $\hat{\mathbf{d}}_t$ is

$$\frac{\partial \mathcal{L}}{\partial \hat{\mathbf{d}}_t} = \frac{\partial \mathcal{L}_{\text{NTP}}}{\partial \hat{\mathbf{d}}_t} + \frac{\partial}{\partial \hat{\mathbf{d}}_t} \left( \frac{\lambda}{L} \sum_{\tau=1}^{n} \hat{\mathbf{d}}_\tau^2 \right) \approx \left( \frac{\partial \mathcal{L}_{\text{NTP}}}{\partial \mathbf{o}_t} \right)^T \frac{\partial \mathbf{o}_t}{\partial \mathbf{d}_t} + \frac{2\lambda}{L} \hat{\mathbf{d}}_t = \left( \frac{\partial \mathcal{L}_{\text{NTP}}}{\partial \mathbf{o}_t} \right)^T . \mathbf{a}_t + \frac{2\lambda}{L} \hat{\mathbf{d}}_t. \tag{15}$$

The approximation ($\approx$) in the above equation reflects the use of Straight Through Estimator (STE) for backpropagation through discrete routing decisions as discussed in Section 3.3. Specifically, $\mathbf{d}_t$ is a discrete routing decision in the forward pass, while the gradients are computed with respect to the continuous routing score $\hat{\mathbf{d}}_t$ during training.

Combining equations 14 and 15:

$$\frac{\partial \mathcal{L}}{\partial \mathbf{W}} = \sum_{t=1}^{n} \frac{\partial \mathcal{L}}{\partial \hat{\mathbf{d}}_t} \frac{\partial \hat{\mathbf{d}}_t}{\partial \mathbf{W}} = \sum_{t=1}^{n} \left[ \frac{\partial \mathcal{L}_{\text{NTP}}}{\partial \mathbf{o}_t} . \mathbf{a}_t \hat{\mathbf{d}}_t (1 - \hat{\mathbf{d}}_t) \mathbf{s}_t^\top + \frac{2\lambda}{L} \hat{\mathbf{d}}_t^2 (1 - \hat{\mathbf{d}}_t) \mathbf{s}_t^\top \right] \tag{16}$$

To interpret the effect of the gradient update on the routing decision, consider the gate logit $\mathbf{z}_t := \mathbf{W}\mathbf{s}_t$, which directly determines the routing score $\hat{\mathbf{d}}_t = \sigma(\mathbf{z}_t)$. Under gradient descent, $\mathbf{W} \leftarrow \mathbf{W} - \eta \frac{\partial \mathcal{L}}{\partial \mathbf{W}}$, the induced change in $\mathbf{z}_t$ is

$$\Delta \mathbf{z}_t = -\eta \left( \frac{\partial \mathcal{L}}{\partial \mathbf{W}} \right) \mathbf{s}_t = -\eta \left[ \left( \frac{\partial \mathcal{L}_{\text{NTP}}}{\partial \mathbf{o}_t} . \mathbf{a}_t \hat{\mathbf{d}}_t (1 - \hat{\mathbf{d}}_t) \right) \|\mathbf{s}_t\|^2 + \left( \frac{2\lambda}{L} \hat{\mathbf{d}}_t^2 (1 - \hat{\mathbf{d}}_t) \right) \|\mathbf{s}_t\|^2 \right] \tag{17}$$

The first term arises from the next token prediction loss (NTP) and provides a context-dependent learning signal that can increase or decrease $\mathbf{z}_t$, enabling the model to selectively activate global attention only when it improves the language modeling objective. The second term arises from the sparsity regularization and is elementwise non-negative, thus driving $\mathbf{z}_t$ toward smaller values, encouraging $\hat{\mathbf{d}}_t \rightarrow 0$ and promoting sparse activation of global attention. For masked tokens (i.e. $\mathbf{a}_t = 0$), the NTP term in Equation (17) vanishes, and the regularization term alone favors solutions with $\mathbf{d}_t \rightarrow 0$, causing the model to never invoke Global Attention. We refer to this behavior as router collapse and discuss it in Section 3.3, along with our strategy for invoking Global Attention across all tokens with Bernoulli sampling (probability $\sim 0.1$).

# B. Extended Related Works

## B.1. Context Length Extrapolation

To mitigate the computational and memory cost of training LLMs on long sequences, several context length extrapolation methods have been proposed (An et al., 2024; Xiao et al., 2024a; Peng et al., 2023; Ding et al., 2024). These approaches extend a model's native context with minimal training by rescaling or reusing positional indices learned during pretraining (Peng et al., 2023; Chen et al., 2023; Liu et al., 2024b), and can be applied directly to pretrained models or used in conjunction with long-context training. For example, Position Interpolation (Chen et al., 2023) linearly scales rotary position embedding (RoPE) indices, while YaRN (Peng et al., 2023) applies uneven frequency interpolation to better preserve high-frequency components.

Another line of work compresses the input using windowed or chunk-based attention (Xiao et al., 2024b; Han et al., 2024), which restricts access to distant context. While hierarchical schemes like Dual-Chunk Attention (DCA) partially reintroduce global information (An et al., 2024; Jin et al., 2024), downstream performance often remains limited, motivating training-based approaches for improved long-context modeling.

## B.2. Dynamic Routing in LLMs

Mixture-of-Experts (MoE) architectures introduced conditional computation as a scalable approach for improving the efficiency of LLM training and inference (Shazeer et al., 2017). MoE consists of multiple expert networks, typically lightweight MLPs, selectively activated for each input through a trainable gating mechanism. While effective at scale, this paradigm generally requires training the entire model from scratch. Several recent works propose training lightweight routing modules on top of frozen backbones, enabling coarse-grained conditional execution such as skipping entire layers based on input difficulty (Heakl et al., 2025).

To achieve more adaptive and fine-grained conditional computation, token-level routing has emerged as a viable alternative (Raposo et al., 2024; Jiang et al., 2024b; Sharma et al., 2025). By operating at the token level, these approaches aim to focus computation on informative tokens while allowing simpler tokens to follow cheaper execution paths. Mixture-of-Depths (MoD) (Raposo et al., 2024) selects a fixed top-$k$ subset of tokens to process at each decoder block using a learned, layer-wise router, while Mixture-of-Recursions (MoR) (Bae et al., 2025) extends this idea to recursive transformers by dynamically determining the number of recursion steps per token. However, both approaches rely on static per-layer compute budgets that route a fixed number of tokens regardless of input complexity or sequence length. In addition, their top-$k$ routing decisions are inherently non-causal. Although auxiliary routers or losses are used to predict routing behavior during inference, they introduce a mismatch between training and inference dynamics, leading to degraded performance. D-LLM (Jiang et al., 2024b) further improves upon deterministic top-$k$ routing by employing Gumbel-Softmax–based routing modules for each layer. Despite more flexible routing, D-LLM, similar to prior token-routing approaches, routes tokens around entire decoder blocks, causing skipped tokens to bypass both attention and MLP layers. While this design improves computational efficiency, it disrupts token interactions and limits representation capacity. These limitations are expected to become more pronounced in long-context settings, where preserving fine-grained token–token interactions over extended sequences is crucial, suggesting that block-level routing is a suboptimal design choice.

## B.3. External Memory-Based Approaches

To address the limitations of fixed context windows, several approaches augment pretrained LLMs with external memory mechanisms to retrieve and incorporate relevant information into the input (Lewis et al., 2020; Chhikara et al., 2025; Borgeaud et al., 2021; Su et al., 2024). Early work in this direction focuses on Retrieval-Augmented Generation (RAG), where the model input is augmented with passages retrieved from an external knowledge corpus (Lewis et al., 2020;

---

**Algorithm 1** *L2A* Kernel Forward Pass

---

**Require:** Matrices $\mathbf{Q}, \mathbf{K}, \mathbf{V} \in \mathbb{R}^{n \times d}$ and non-zero query indices $\mathbf{Q}_{idx} \in \mathbb{R}^n$

1: Consolidate active (non-zero) queries into $\mathbf{Q}_c \in \mathbb{R}^{n_c \times d}$ using indices $\mathbf{Q}_{idx}$.

2: $\mathbf{Q}_c$ tiled into $T_q = n_c/B_q$ blocks ($\mathbf{Q}_{c,1}, \ldots, \mathbf{Q}_{c,T_q}$ of size $B_q \times d$), $\mathbf{K}, \mathbf{V}$ into $T_k = n/B_k$ blocks ($\mathbf{K}_1, \ldots, \mathbf{K}_{T_k}$, $\mathbf{V}_1, \ldots, \mathbf{V}_{T_k}$, of size $B_k \times d$), and $\mathbf{Q}_{idx}$ into $T_q = n/B_q$ blocks ($\mathbf{Q}_{idx,1}, \ldots, \mathbf{Q}_{idx,T_q}$ of size $B_q$)

3: Initialize $\mathbf{O} \leftarrow \mathbf{0}_{n \times d}$. Partition $\mathbf{O}_c \in \mathbb{R}^{n_c \times d}$ into $T_q$ tiles ($\mathbf{O}_{c,i}, \ldots, \mathbf{O}_{c,T_q}$ of size $B_q \times d$), and divide the logsumexp $L_c$ into $T_q$ blocks ($L_{c,i}, \ldots, L_{c,T_q}$ of size $B_q$).

4: **for** $1 \leq i \leq T_q$ **do**

5:      Load $\mathbf{Q}_{c,i}$ and $\mathbf{Q}_{idx,i}$ tiles from HBM to on-chip SRAM.

6:      $max\_k\_blocks = max(\mathbf{Q}_{idx,i})/B_k$

7:      Initialize $\mathbf{O}_{c,i}^{(0)} = (0)_{B_q \times d} \in \mathbb{R}^{B_q \times d}, \ell_{c,i}^{(0)} = (0)_{B_q} \in \mathbb{R}^{B_q}, m_{c,i}^{(0)} = (-\infty)_{B_q} \in \mathbb{R}^{B_q}$.

8:      **for** $1 \leq j \leq max\_k\_blocks$ **do**

9:          Load $\mathbf{K}_j, \mathbf{V}_j$ tiles from HBM to on-chip SRAM with ids $\mathbf{K}_{idx}$.

10:         $\mathbf{S}_i^{(j)} = (\mathbf{Q}_{c,i} \mathbf{K}_j^T)/\sqrt{d} \in \mathbb{R}^{B_q \times B_k}$ with element-wise causal mask $\mathbf{Q}_{idx,i} >= \mathbf{K}_{idx}$.

11:         $m_{c,i}^{(j)} = \max(m_{c,i}^{(j-1)}, \text{rowmax}(\mathbf{S}_i^{(j)})) \in \mathbb{R}^{B_q}$

12:         $\tilde{\mathbf{P}}_i^{(j)} = \exp(\mathbf{S}_i^{(j)} - m_{c,i}^{(j)}) \in \mathbb{R}^{B_q \times B_k}$

13:         $\ell_{c,i}^{(j)} = e^{m_{c,i}^{(j-1)} - m_{c,i}^{(j)}} \ell_{c,i}^{(j-1)} + \text{rowsum}(\tilde{\mathbf{P}}_i^{(j)}) \in \mathbb{R}^{B_q}$.

14:         $\mathbf{O}_{c,i}^{(j)} = \text{diag}(e^{m_{c,i}^{(j-1)} - m_{c,i}^{(j)}})^{-1} \mathbf{O}_{c,i}^{(j-1)} + \tilde{\mathbf{P}}_i^{(j)} \mathbf{V}_j$.

15:      **end for**

16:      $\mathbf{O}_{c,i} = \text{diag}(\ell_{c,i}^{(T_q)})^{-1} \mathbf{O}_{c,i}^{(T_q)}$.

17:      $L_{c,i} = m_{c,i}^{(T_q)} + \log(\ell_{c,i}^{(T_q)})$.

18:      Write $\mathbf{O}_{c,i}$ and $L_{c,i}$ to HBM as the $i$-th tiles of $\mathbf{O}_c$ and $L_c$, respectively.

19: **end for**

20: Save $\mathbf{Q}_c, \mathbf{K}, \mathbf{V}, \mathbf{O}_c, L_c, \mathbf{Q}_{idx}$ for backward.

21: Scatter $\mathbf{O}_c$ to $\mathbf{O}$ using $\mathbf{Q}_{idx}$ (i.e., $\mathbf{O}[\mathbf{Q}_{idx}] \leftarrow \mathbf{O}_c$).

22: **return** $\mathbf{O}$ and $L_c$

---

Borgeaud et al., 2021). RAG provides an efficient complement to long-context inference, but its effectiveness is contingent upon the quality and relevance of the retrieved information (Asai et al., 2024; Yan et al., 2024). This can be mitigated through nearest-neighbor based retrieval (Borgeaud et al., 2021), self-corrective and adaptive retrieval strategies (Yan et al., 2024; Jeong et al., 2024) and context-based graphical external memory representations (Chhikara et al., 2025). Despite these advances, external memory-based approaches often lag behind models that directly operate over long contexts (Li et al., 2024b; Zhou et al., 2025). This gap is commonly attributed to the limited ability of such approaches to discern complex and implicit dependencies and to attend to relevant context. In contrast, our approach treats long-term memory access as a native operation within the model, enabling conditional global attention to perform context-based retrieval directly over the input sequence.

## C. *L2A* **Kernel Forward Pass**

Algorithm 1 outlines the forward pass of the *L2A* kernel, which extends FlashAttention-2 to support token-level conditional Global Attention. Active queries are first compacted into a contiguous buffer $\mathbf{Q}_c$ using the index mapping $\mathbf{Q}_{idx}$, enabling efficient tiled computation. The kernel processes $\mathbf{Q}_c$, $\mathbf{K}$, and $\mathbf{V}$ in blocks streamed from high-bandwidth memory into on-chip SRAM, and for each query tile iterates only over key–value tiles that lie within its causal range as determined by $\mathbf{Q}_{idx}$. Attention scores are accumulated using an online log-sum-exp scheme to ensure numerical stability without materializing the full attention matrix. After processing all relevant tiles, the compact outputs $\mathbf{O}_c$ and normalization statistics $L_c$ are written back to memory and scattered to reconstruct the full output $\mathbf{O}$. This design preserves the efficiency and numerical stability of FA-2 while avoiding unnecessary computation for inactive queries.

# D. Additional Results

## D.1. *L2A* Kernel Speedups

We benchmark the forward and backward pass of *L2A* kernel on NVIDIA H200 GPUs and compare it against FlashAttention-2. As shown in Figure 9, our custom kernel achieves 1.6-35× and 1.08-45 × speedups for the forward and the backward pass across varying sparsity levels, where sparsity denotes the fraction of queries for which Global Attention is skipped.

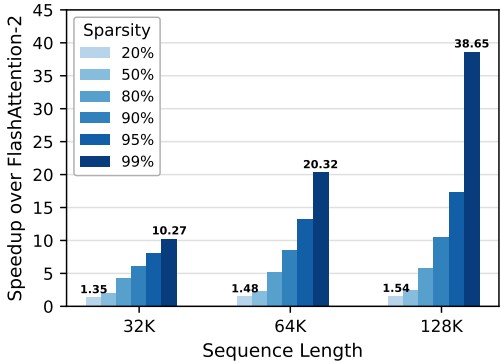 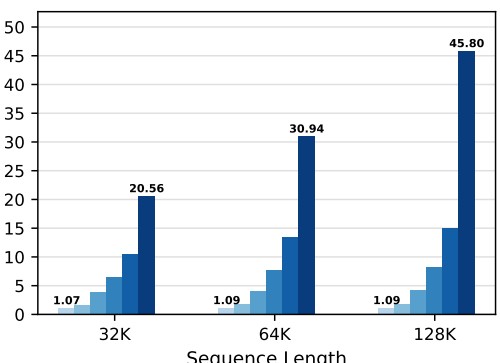

*Figure 9. L2A* kernel speedups over FlashAttention-2 for the forward (left) and backward (right) pass across varying sparsity levels and sequence lengths. Qwen-2.5 7B configuration is used (28 attention heads and a head dimension of 128).

Note that the kernel-level speedups reported in Figure 9 include both the sparse Attention kernel and the preprocessing stage required for dynamic query compaction and index construction. We provide a runtime breakdown in Table 4 to highlight the impact of preprocessing overhead at shorter context lengths. At 4K context length, preprocessing accounts for 45–66% of the total forward-pass runtime, limiting overall speedup at low sparsity levels. However, as sparsity increases, the reduction in Attention computation compensates for this overhead, resulting in net speedups over FlashAttention-2. In practice, our L2A models achieve 90+% sparsity on the short-context tasks reported in the paper, where our kernels are already faster than FlashAttention-2 even at 4K context length. More importantly, the primary goal of L2A is to enable efficient long-context training, where Attention cost remains a major bottleneck, and the benefits of sparsity become substantially larger.

*Table 4.* Forward-pass runtime breakdown of *L2A* at short context lengths. While preprocessing overhead is significant at small sequence lengths, higher sparsity levels compensate for this cost and yield net speedups over FlashAttention. Preprocessing (%) denotes the fraction of runtime spent on routing and index construction.

| Seq Len | FlashAttn (ms) | Sparsity | L2A Total (ms) | Preprocessing (%) | Speedup |
|---|---|---|---|---|---|
| 4096 | 0.41 | 0.20 | 0.59 | 45.40 | 0.70× |
| | | 0.50 | 0.49 | 56.40 | 0.83× |
| | | 0.80 | 0.40 | 63.10 | 1.02× |
| | | 0.90 | 0.37 | 65.80 | 1.10× |
| | | 0.95 | 0.36 | 66.31 | 1.13× |
| 8192 | 1.38 | 0.20 | 1.52 | 32.23 | 0.90× |
| | | 0.50 | 1.14 | 36.07 | 1.21× |
| | | 0.80 | 0.75 | 50.16 | 1.84× |
| | | 0.90 | 0.63 | 58.72 | 2.19× |
| | | 0.95 | 0.59 | 65.10 | 2.33× |

# E. Additional Analysis of Native Sparse Attention (NSA)

As mentioned in Section 4.1, the original NSA work (Yuan et al., 2025) does not provide an official kernel implementation. Therefore, we rely on a publicly available open-source implementation (Li, 2025) to evaluate NSA in our experiments. To verify the correctness of this implementation, we reproduce the Needle-in-a-Haystack retrieval results reported in the original paper at a sequence length of 64K. We obtain perfect retrieval accuracy, matching the results reported in the paper, indicating that the implementation faithfully realizes the NSA algorithm.

*Table 5.* Short-context performance of a standard Transformer and *L2A* when trained from scratch using the Qwen2.5-1.5B architecture.

| Task | Transformer | L2A |
|------|-------------|-----|
| BoolQ | 60.86 | 55.66 |
| CommonSenseQA | 19.25 | 19.16 |
| MMLU | 25.80 | 25.14 |
| PIQA | 71.76 | 71.33 |
| SWDE | 65.98 | 65.89 |
| Winogrande | 58.56 | 59.04 |
| ARC-C | 24.83 | 25.17 |
| ARC-E | 25.51 | 24.41 |
| Average | 44.06 | 43.23 |

*Table 6.* Average long-context performance of a standard Transformer and *L2A* trained from scratch. Despite achieving ∼60% sparsity, *L2A* remains competitive with the Transformer across context lengths while improving training and inference efficiency.

| Model | 8k | 16k | 32k | 64k | 128k |
|-------|-----|------|------|------|-------|
| Transformer | 15.0 | 14.0 | 12.4 | 11.0 | 10.1 |
| L2A | 17.0 | 15.7 | 14.4 | 13.0 | 11.8 |

We then analyze the kernel-level performance of this implementation relative to FlashAttention-2 (Dao, 2023) across varying sequence lengths. As shown in Figure 10, the evaluated NSA kernel is consistently slower than FlashAttention-2 for both the forward and backward passes. We emphasize that this observation does not contradict the claims of the original NSA paper. Specifically, the original design proposes executing the compression, top-$n$ block selection, and Sliding Window Attention (SWA) components in parallel, whereas the available open-source kernel executes these stages sequentially. The absence of parallel execution in this implementation is likely the primary contributor to the observed performance gap.

In all experiments, we follow the hyperparameters described in the original NSA paper as closely as possible. The only deviation is the choice of selected block count $n$, and we find the setting in the original paper ($n = 16$) to result in degraded performance in our setup. We therefore increase $n$ to 128 in all NSA experiments. Notably, as shown in Figure 4, even with this substantially larger $n$, which is more favorable to NSA than the original paper setting, *L2A* significantly outperforms NSA across all evaluated tasks. This highlights the advantage of learning when to invoke global attention on a per-token basis, as opposed to relying on a fixed long-range context budget. Although NSA learns which tokens to attend within the context, the amount of long-context capacity is predetermined and applied uniformly across tokens.

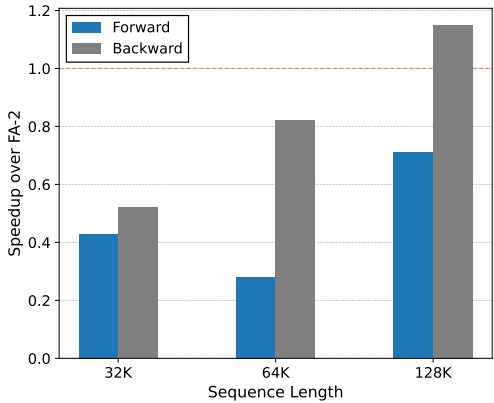

*Figure 10.* Native Sparse Attention (NSA) kernel (Li, 2025) speedups over FlashAttention-2 for the forward and backward pass across various sequence lengths. All experiments use the Qwen 2.5 7B configuration with 28 attention heads and a head dimension of 128.

*Table 7.* Average long-context performance across context lengths with Qwen2.5-7B as base LLM. Applying sparsity-driven layer pruning to *L2A* removes nearly 40% of Global Attention layers while maintaining performance close to both CLP and the *L2A* model.

| Model | 8k | 16k | 32k | 64k | 128k |
|---|---|---|---|---|---|
| CLP-7B | 56.0 | 52.1 | 49.0 | 45.4 | 40.7 |
| L2A | 53.8 | 51.4 | 48.3 | 45.6 | 41.6 |
| L2A [Pruned] | 52.9 | 50.2 | 46.7 | 45.2 | 40.6 |

### E.1. Ablation Study

#### E.1.1. PRETRAINING L2A FROM SCRATCH.

We focus on developing long-context methods for off-the-shelf Transformer models, motivated by the high cost of pretraining, which often requires trillions of tokens to obtain a strong model (Qwen et al., 2025). To further evaluate the method, we also consider training from scratch and train both a standard Transformer and L2A using the Qwen2.5-1.5B architecture. Models are trained on 100B tokens from DCLM (Li et al., 2024a) at 4K context, followed by continued pretraining at 128K, and evaluated on short-context and long-context benchmarks using the same setup as in the main paper. L2A performs within 1 point of the Transformer on short-context tasks (Table 5) while remaining competitive on long-context benchmarks as shown in Table 6. It achieves an average sparsity of 60% across tasks, resulting in 1.25x training speedup and 1.6x improvement in average time-to-first-token during inference at 128K context. Overall, these results highlight L2A's efficiency gains even when trained from scratch, despite no additional parameter tuning.

#### E.1.2. ABLATION ON THE REGULARIZATION COEFFICIENT.

We next study the effect of the regularization coefficient $\lambda$ in the training objective (Equation (6)) on *L2A* 's performance. As expected, increasing $\lambda$ induces higher sparsity, leading to greater reliance on Local Attention and, in turn, degraded long-context performance (Figure 11). However, tuning $\lambda$ is challenging because its values do not map directly to specific sparsity levels. To address this, we instead vary the sigmoid threshold in Equation (9), which enables direct control over Global Attention sparsity at test time.

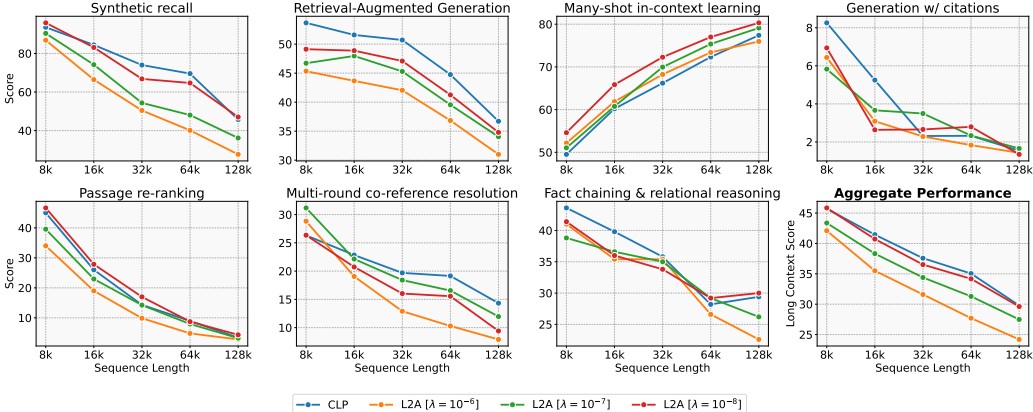

*Figure 11.* Ablation on the regularization coefficient $\lambda$. Increasing $\lambda$ induces higher sparsity, leading to greater reliance on Local Attention and degraded long-context performance.

#### E.1.3. TEST-TIME COMPUTE SCALING THROUGH SIGMOID THRESHOLD.

In all experiments, we apply a fixed threshold of $0.5$ to the sigmoid output of the routing module to obtain discrete routing decisions, as defined in Equation 3. To examine test-time compute scaling, we vary this threshold at inference to control the sparsity of global attention. For a range of threshold values as shown in Figure 12, increasing the threshold leads to higher sparsity and lower time-to-first-token, at the cost of reduced aggregate performance, while lower thresholds recover performance at increased compute cost. This enables flexible test-time control over the compute–accuracy trade-off, allowing inference cost to be modulated without retraining.

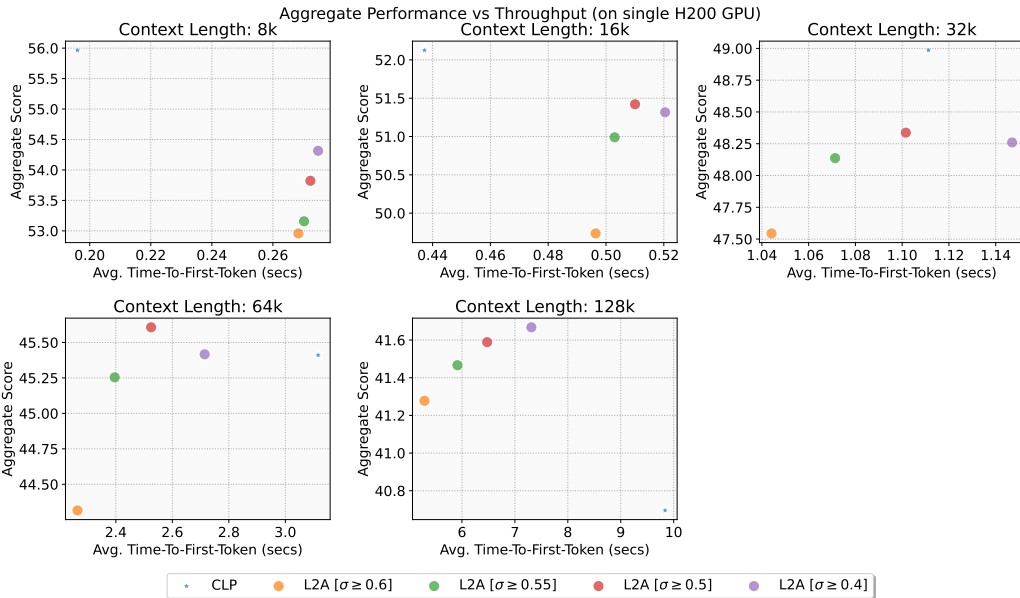

*Figure 12.* Scaling test time compute for *L2A* by varying the threshold of the Routing module (Equation (3)). Increasing the threshold leads to higher sparsity and lower time-to-first-token, at the cost of reduced aggregate performance.

### E.1.4. COMPARING FULL-MODEL FINE-TUNING WITH ATTENTION AND LAYERNORM-ONLY TRAINING

As described in Section 4, we restrict training to the Attention and LayerNorm layers while freezing the feed-forward (FFN) layers. This design choice is motivated by prior work showing that a large fraction of the model's pretrained parametric knowledge is encoded in FFNs (Geva et al., 2021), while attention layers primarily focus on contextual routing and composition. By preserving FFNs, we aim to retain the pretrained capabilities of the Base LLM while adapting to long-context modeling. As shown in Table 11, this strategy consistently outperforms full-model fine-tuning.

## F. Implementation Details

### F.1. Training Hyperparameters

All models are trained using the AdamW optimizer with an initial learning rate of $5 \times 10^{-5}$, 5% warmup steps, cosine learning rate scheduling, and gradient clipping with a norm of 1. We use a batch size of 6M tokens for the Qwen 2.5 7B-based and Qwen3 8B-based models, and 4M tokens for the Qwen 2.5 1.5B model. All experiments are conducted with a sequence length of 128K tokens using BF16 training. We train for 25B tokens the Qwen2.5 7B- and Qwen3 8B-based models, while for the smaller Qwen2.5 1.5B-based models we train on 16.7B tokens.

### F.2. Long-context Benchmarks

We consider a suite of long-context evaluation tasks drawn from HELMET (Yen et al., 2025), BabiLong (Kuratov et al., 2024), and MRCR (Vodrahalli et al., 2024), which we describe below.

- *Synthetic Recall*: These tasks are variants of the needle-in-a-haystack setting (Kamradt, 2024), in which the model must retrieve a critical piece of information (the "needle") from a long sequence of irrelevant or distracting tokens (the "haystack"). In addition to retrieval, these variants evaluate the model's ability to perform multi-hop tracing and information aggregation across the context. Our Recall evaluation consists of these tasks from HELMET (Yen et al., 2025): JSON Key–Value, NIAH Multi-Key (MK) Needle, NIAH Multi-Key (MK) UUID, and NIAH Multi-Value (MV).

- *Multi-round Co-reference Resolution (MRCR)*: MRCR (Vodrahalli et al., 2024) measures an LLM's ability to resolve multiple reference needles distributed across a long context. The model is presented with a multi-turn, synthetically generated conversation in which the user issues repeated requests for a piece of writing on a given topic. Embedded

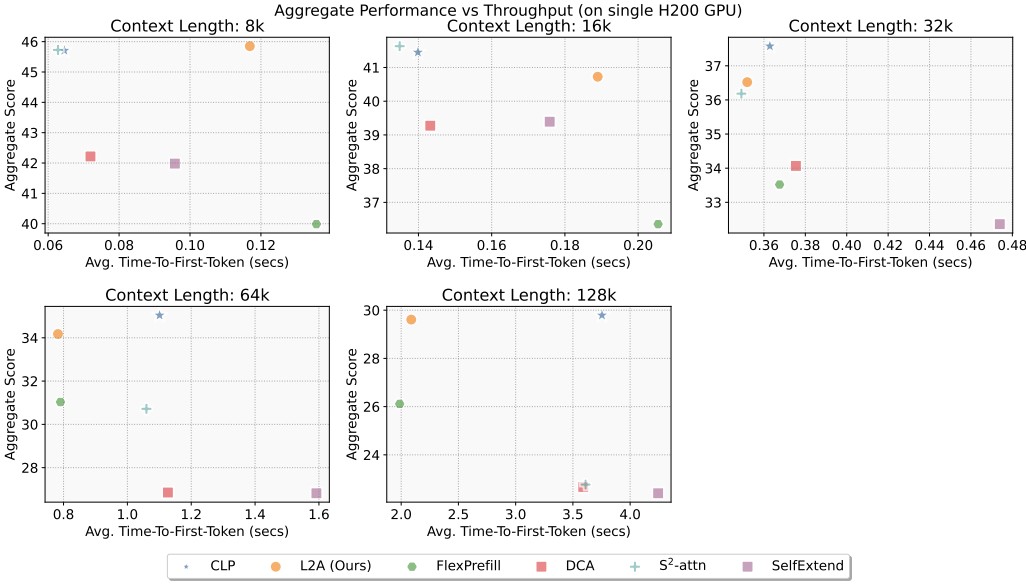

*Figure 13.* Time-to-first-token (TTFT) for Qwen2.5 1.5B model for various sequence lengths. *L2A* becomes progressively faster as the context length increases, highlighting the need for conditional Global Attention.

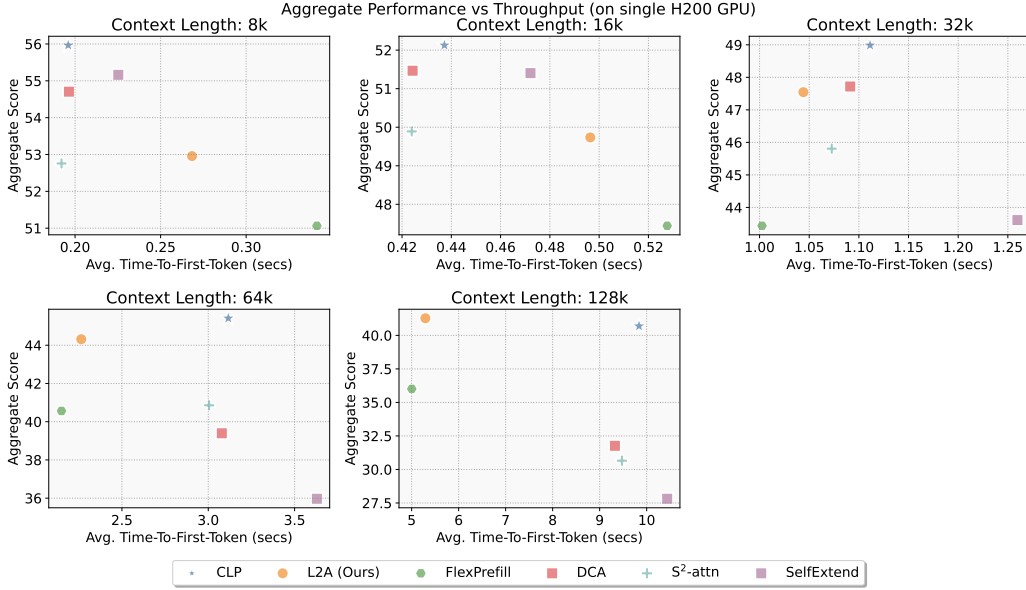

*Figure 14.* Time-to-first-token (TTFT) for Qwen2.5 7B model for various sequence lengths. *L2A* becomes progressively faster as the context length increases, highlighting the need for conditional Global Attention.

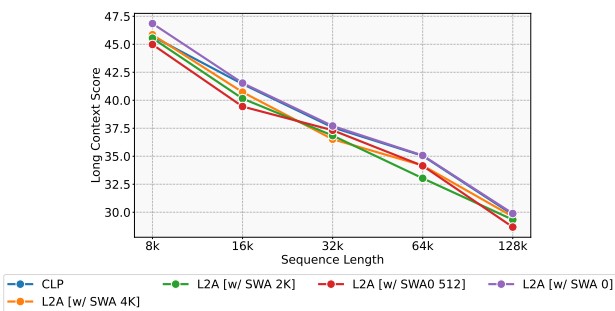 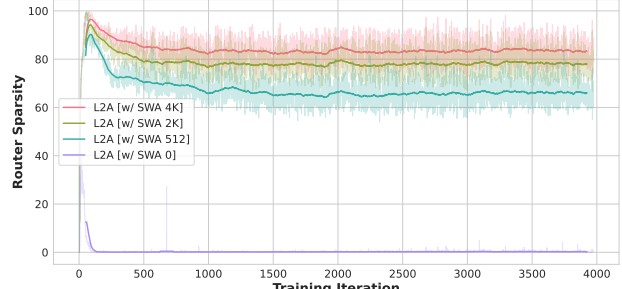

*Figure 15.* Aggregate long-context performance for *L2A* variants with varying SWA window sizes. Performance remains similar, while the Router compensates for smaller SWA windows by reducing sparsity in Global Attention invocations.

*Figure 16.* Sparsity levels for *L2A* variants with different SWA sizes. As the short-term local context shrinks, *L2A* increasingly relies on long-term global memory. The SWA-0 configuration exhibits 0% sparsity during training.

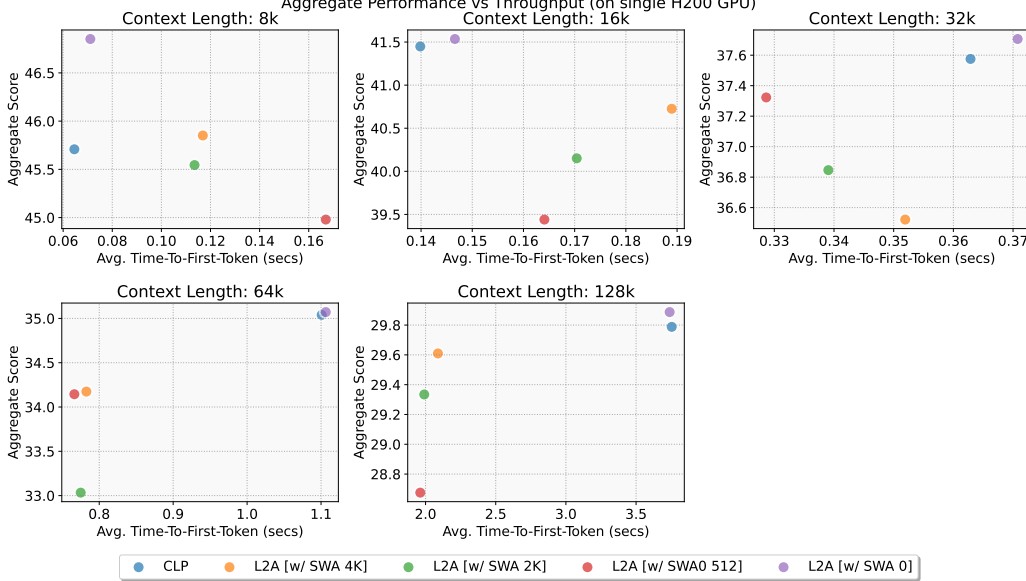

*Figure 17.* Time-to-first token (TTFT) for *L2A* variants with varying SWA context window from 0-4K. These results reveal a clear tradeoff: as the short-term local context shrinks, *L2A* increasingly relies on long-term global memory resulting into higher TTFT as shown.

within the conversation are a few identical requests, and the model is ultimately prompted to return a specific instance. Task difficulty increases with both the context length and the number of needles.

- *Retrieval Augmented Generation (RAG)*: These tasks involve open-domain question answering, where the model is provided with a gold passage containing the answer, interleaved with numerous distractor passages retrieved from a large corpus. The model is required to answer the question using the provided passages. We evaluate on the following datasets from HELMET (Yen et al., 2025): Natural Questions, TriviaQA, PopQA, and HotpotQA.

- *Many-shot In-context Learning (ICL)*: ICL evaluates an LLM's ability to acquire new skills from a small number of examples provided in the prompt. In this setting, the model learns to classify inputs into different concepts based on several in-context demonstrations. We evaluate ICL performance using the following datasets from HELMET: TREC Coarse, TREC Fine, NLU, BANKING77, and CLINIC150.

- *Generation with Citations (Cite)*: LLMs are benchmarked on a realistic question-answering setting that requires generating responses with correct attributions (Bohnet et al., 2023). Given multi-faceted questions and a set of relevant passages, models are tasked with producing long-form answers while citing supporting passage identifiers (Gao et al., 2023). This evaluates the model's ability to reason over the provided context and its adherence to citation instructions.

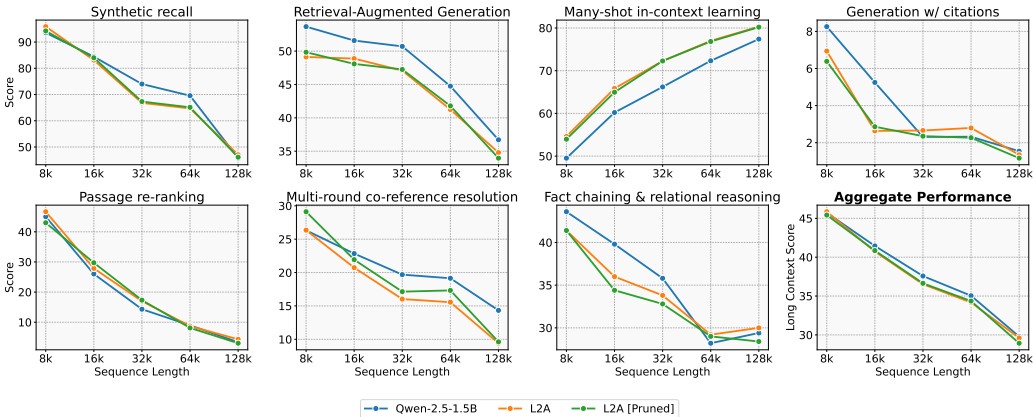

*Figure 18.* L2A [Pruned] is the model obtained by pruning Global Attention Modules that are more than 95% sparse—that is, layers where the Global Attention Module is not invoked for 95% of tokens. The figure shows minimal loss in long-context performance compared to L2A.

We consider the ASQA (Stelmakh et al., 2023) and QAMPARI (Amouyal et al., 2023) subsets, and the outputs are evaluated for both answer correctness and citation quality.

- *Passage Re-ranking (Re-rank)*: We evaluate passage re-ranking performance using the MS MARCO benchmark (Bajaj et al., 2018) and annotated datasets from the TREC Passage Re-ranking challenge (Craswell et al., 2025). Each instance consists of a query and a set of candidate passages labeled by relevance. For a given input length $L$, we select $k$ passages as context, balance label distributions, and randomize passage order to mitigate positional bias, generating three permutations per instance. Performance is measured using NDCG@10, which balances evaluation fidelity with the computational cost of inference.

- *Fact Chaining & Relational Reasoning (FCRR)*: We evaluate long-context fact chaining and relational reasoning using the QA1–QA5 tasks from the BabiLong benchmark (Kuratov et al., 2024). These tasks are designed to assess progressively more challenging reasoning capabilities, ranging from single-fact retrieval (QA1) to multi-hop fact chaining (QA2 and QA3) and relational reasoning over two- and three-argument relations (QA4 and QA5). Relevant facts are sparsely distributed across long sequences containing a large number of distractor tokens, requiring models to selectively retrieve and compose information over extended contexts. Performance on these tasks reflects a model's ability to handle long-range dependencies, multi-step reasoning, and structured relational inference under long-context noise.

*Table 8.* Task-wise performance comparison of *L2A* and baseline methods across long-context benchmarks on Qwen 2.5 7B model. *L2A* remains close to CLP performance while outperforming baselines on most tasks. All scores are reported on a 0–100 scale.

| Task | Model | Context Length | | | | |
|---|---|---|---|---|---|---|
| | | **8k** | **16k** | **32k** | **64k** | **128k** |
| Synthetic recall | CLP | **99.1** | 98.5 | **98.7** | **95.4** | 76.4 |
| | L2A (Ours) | 98.7 | 97.3 | 96.5 | 93.9 | **79.2** |
| | FlexPrefill | 98.1 | 96.5 | 91.8 | 88.9 | 69.8 |
| | DCA | 98.6 | 96.7 | 94.0 | 71.9 | 43.9 |
| | S$^2$-attn | **99.1** | **98.8** | 97.6 | 74.6 | 29.0 |
| | SelfExtend | 98.6 | 96.4 | 78.2 | 55.3 | 25.9 |
| | NSA | 97.9 | 75.0 | 29.1 | 13.8 | 6.2 |
| | Base | 98.6 | 97.1 | 95.9 | 45.1 | 13.9 |
| Retrieval-Augmented Generation | CLP | **64.7** | **62.1** | **59.9** | **57.3** | **50.0** |
| | L2A (Ours) | 62.0 | 60.4 | 57.7 | 55.1 | **50.0** |
| | FlexPrefill | 59.9 | 58.9 | 55.0 | 53.9 | 49.7 |
| | DCA | 62.6 | 60.3 | 53.9 | 48.8 | 43.1 |
| | S$^2$-attn | 63.2 | 60.3 | 56.9 | 53.6 | 44.1 |
| | SelfExtend | 62.9 | 60.5 | 52.5 | 46.8 | 38.1 |
| | NSA | 61.7 | 55.1 | 48.9 | 45.0 | 39.2 |
| | Base | 62.5 | 60.1 | 54.3 | 43.7 | 27.5 |
| Many-shot in-context learning | CLP | 73.6 | 79.4 | 81.7 | 84.5 | 87.4 |
| | L2A (Ours) | 71.4 | **79.6** | **83.5** | **87.0** | **88.6** |
| | FlexPrefill | **74.2** | 78.5 | 82.3 | 84.2 | 85.2 |
| | DCA | 70.0 | 75.6 | 78.7 | 81.2 | 82.4 |
| | S$^2$-attn | 61.8 | 67.0 | 71.5 | 76.7 | 77.0 |
| | SelfExtend | 70.2 | 75.0 | 78.0 | 81.2 | 81.8 |
| | NSA | 66.6 | 71.0 | 74.2 | 78.2 | 78.6 |
| | Base | 70.1 | 75.1 | 78.4 | 78.5 | 71.5 |
| Generation w/ citations | CLP | **24.2** | **18.3** | 8.6 | **4.8** | **2.8** |
| | L2A (Ours) | 17.2 | 10.5 | 8.5 | 4.1 | 1.9 |
| | FlexPrefill | 13.3 | 6.8 | 5.1 | 3.6 | 2.4 |
| | DCA | 21.9 | 15.5 | **9.5** | 2.1 | 1.2 |
| | S$^2$-attn | 20.9 | 14.9 | 8.4 | 2.9 | 2.5 |
| | SelfExtend | 22.5 | 16.3 | 5.9 | 2.5 | 2.3 |
| | NSA | 15.3 | 6.4 | 2.8 | 2.1 | 1.5 |
| | Base | 21.7 | 16.3 | 8.2 | 2.2 | 1.3 |
| Passage re-ranking | CLP | 48.1 | 32.5 | **25.1** | **18.1** | 8.4 |
| | L2A (Ours) | 44.0 | 26.8 | 20.1 | 12.5 | **9.6** |
| | FlexPrefill | **50.5** | 30.8 | 19.4 | 9.2 | 5.0 |
| | DCA | 46.3 | **33.5** | 23.2 | 9.6 | 1.8 |
| | S$^2$-attn | 46.7 | 32.7 | 24.2 | 14.0 | 3.8 |
| | SelfExtend | 47.3 | 33.2 | 18.1 | 6.8 | 0.6 |
| | NSA | 42.0 | 9.4 | 4.5 | 0.3 | 0.2 |
| | Base | 46.3 | 33.2 | 22.7 | 1.5 | 0.0 |
| Multi-round co-reference resolution | CLP | 34.6 | 30.4 | 28.9 | **26.3** | **21.3** |
| | L2A (Ours) | 32.9 | 31.5 | 25.8 | 24.2 | 19.3 |
| | FlexPrefill | 29.7 | 27.0 | 28.3 | 24.5 | 17.0 |
| | DCA | 32.0 | 30.9 | 29.9 | 22.4 | 14.8 |
| | S$^2$-attn | 32.5 | **31.8** | 26.9 | 25.9 | 19.0 |
| | SelfExtend | 32.4 | 30.7 | **30.8** | 21.0 | 11.4 |
| | NSA | **35.0** | 26.1 | 16.6 | 13.7 | 8.7 |
| | Base | 32.4 | 31.5 | 29.6 | 25.8 | 11.6 |
| Fact chaining & relational reasoning | CLP | 47.4 | 43.6 | 40.0 | 31.4 | 38.6 |
| | L2A (Ours) | 44.6 | 42.2 | 40.6 | 33.4 | **40.4** |
| | FlexPrefill | 47.6 | 42.6 | 37.8 | 33.4 | 35.8 |
| | DCA | 51.6 | **47.8** | **44.8** | 39.8 | 35.2 |
| | S$^2$-attn | 45.2 | 43.8 | 35.2 | 38.2 | 39.2 |
| | SelfExtend | **52.2** | 47.6 | 41.8 | 38.2 | 34.6 |
| | NSA | 42.2 | 39.2 | 31.2 | 24.0 | 17.6 |
| | Base | 51.6 | **47.8** | 43.0 | **41.8** | 21.4 |
| Average across tasks | CLP | **56.0** | **52.1** | **49.0** | **45.4** | 40.7 |
| | L2A (Ours) | 53.0 | 49.7 | 47.5 | 44.3 | **41.3** |
| | FlexPrefill | 53.3 | 48.7 | 45.7 | 42.5 | 37.8 |
| | DCA | 54.7 | 51.5 | 47.7 | 39.4 | 31.8 |
| | S$^2$-attn | 52.8 | 49.9 | 45.8 | 40.9 | 30.7 |
| | SelfExtend | 55.2 | 51.4 | 43.6 | 36.0 | 27.8 |
| | NSA | 51.5 | 40.3 | 29.6 | 25.3 | 21.7 |
| | Base | 54.7 | 51.6 | 47.4 | 34.1 | 21.0 |

*Table 9.* Task-wise performance comparison of *L2A* and baseline methods across long-context benchmarks on Qwen 2.5 1.5B model. *L2A* remains close to CLP performance while outperforming baselines on most tasks. All scores are reported on a 0–100 scale.

| Task | Model | Context Length | | | | |
|---|---|---|---|---|---|---|
| | | 8k | 16k | 32k | 64k | 128k |
| Synthetic recall | CLP | 93.6 | **84.4** | **74.0** | **69.6** | 45.8 |
| | L2A (Ours) | **95.9** | 83.1 | 66.8 | 64.7 | **47.1** |
| | FlexPrefill | 90.4 | 77.4 | 60.2 | 54.9 | 31.9 |
| | DCA | 85.9 | 81.1 | 60.8 | 32.7 | 18.3 |
| | $S^2$-attn | 93.5 | 81.1 | 64.8 | 41.2 | 16.3 |
| | SelfExtend | 85.1 | 80.7 | 52.1 | 33.9 | 21.7 |
| | NSA | 76.2 | 29.7 | 14.2 | 9.3 | 4.1 |
| | Base | 85.4 | 80.1 | 64.5 | 19.4 | 3.4 |
| Retrieval-Augmented Generation | CLP | **53.7** | **51.6** | **50.7** | **44.8** | **36.7** |
| | L2A (Ours) | 49.1 | 48.9 | 47.1 | 41.2 | 34.8 |
| | FlexPrefill | 52.0 | 51.0 | 47.2 | 43.0 | 35.8 |
| | DCA | 49.7 | 46.0 | 42.2 | 36.3 | 28.8 |
| | $S^2$-attn | 51.8 | 50.6 | 48.6 | 39.6 | 27.4 |
| | SelfExtend | 49.4 | 45.9 | 39.8 | 33.6 | 27.4 |
| | NSA | 48.5 | 44.0 | 39.0 | 32.3 | 30.2 |
| | Base | 49.8 | 46.1 | 41.8 | 30.2 | 22.8 |
| Many-shot in-context learning | CLP | 49.5 | 60.2 | 66.2 | 72.3 | 77.4 |
| | L2A (Ours) | 54.6 | **65.8** | **72.3** | **77.0** | **80.3** |
| | FlexPrefill | 39.2 | 53.0 | 63.0 | 68.9 | 72.0 |
| | DCA | 54.7 | 64.8 | 69.8 | 74.2 | 76.3 |
| | $S^2$-attn | 54.2 | 62.4 | 68.8 | 73.8 | 72.8 |
| | SelfExtend | 54.4 | 65.2 | 70.2 | 73.6 | 74.4 |
| | NSA | **61.5** | 65.0 | 65.4 | 65.3 | 65.0 |
| | Base | 54.5 | 65.2 | 69.4 | 68.1 | 43.6 |
| Generation w/ citations | CLP | 8.3 | 5.2 | 2.3 | 2.3 | 1.5 |
| | L2A (Ours) | 6.9 | 2.6 | 2.7 | 2.8 | 1.3 |
| | FlexPrefill | 5.5 | 3.1 | 3.3 | 1.9 | **1.6** |
| | DCA | 7.9 | **5.3** | 3.2 | 1.4 | 0.9 |
| | $S^2$-attn | **8.9** | 3.8 | 3.5 | **3.0** | 1.4 |
| | SelfExtend | 7.4 | 5.0 | 2.9 | 1.4 | 1.0 |
| | NSA | 7.1 | 3.4 | 2.6 | 1.6 | 0.8 |
| | Base | 8.1 | 4.9 | **4.4** | 0.7 | 0.9 |
| Passage re-ranking | CLP | 45.0 | 26.0 | 14.3 | **9.0** | 3.4 |
| | L2A (Ours) | **46.7** | 27.9 | **17.0** | 8.7 | **4.3** |
| | FlexPrefill | 25.7 | 13.0 | 8.4 | 3.5 | 2.0 |
| | DCA | 33.6 | 20.0 | 9.2 | 1.7 | 0.0 |
| | $S^2$-attn | 41.4 | **31.1** | 12.3 | 8.0 | 2.2 |
| | SelfExtend | 34.4 | 20.3 | 8.9 | 0.4 | 0.0 |
| | NSA | 32.8 | 16.3 | 10.0 | 1.5 | 1.4 |
| | Base | 33.5 | 21.6 | 10.0 | 4.3 | 0.0 |
| Multi-round co-reference resolution | CLP | 26.3 | 22.8 | **19.7** | 19.1 | **14.3** |
| | L2A (Ours) | 26.4 | 20.7 | 16.0 | 15.6 | 9.4 |
| | FlexPrefill | 24.0 | 19.4 | 16.5 | 13.0 | 12.1 |
| | DCA | 26.7 | 22.8 | 17.6 | 13.9 | 11.0 |
| | $S^2$-attn | **26.8** | 22.2 | 19.0 | 17.8 | 12.8 |
| | SelfExtend | 26.3 | **23.1** | 18.6 | 18.7 | 8.9 |
| | NSA | 25.6 | 18.7 | 8.3 | 9.7 | 6.1 |
| | Base | 26.4 | 22.7 | 18.0 | 15.4 | 5.6 |
| Fact chaining & relational reasoning | CLP | **43.6** | 39.8 | 35.8 | 28.2 | 29.4 |
| | L2A (Ours) | 41.4 | 36.0 | 33.8 | 29.2 | **30.0** |
| | FlexPrefill | 43.2 | 37.6 | 36.0 | **32.0** | 27.4 |
| | DCA | 37.0 | 35.0 | 35.6 | 27.8 | 23.4 |
| | $S^2$-attn | 43.4 | **40.2** | **36.2** | 31.6 | 26.4 |
| | SelfExtend | 37.0 | 35.6 | 34.0 | 26.2 | 23.4 |
| | NSA | 40.4 | 30.6 | 23.0 | 18.0 | 11.0 |
| | Base | 38.2 | 35.2 | 31.0 | 24.0 | 13.0 |
| Average across tasks | CLP | 45.7 | 41.4 | **37.6** | 35.0 | **29.8** |
| | L2A (Ours) | **45.9** | 40.7 | 36.5 | 34.2 | 29.6 |
| | FlexPrefill | 40.0 | 36.4 | 33.5 | 31.0 | 26.1 |
| | DCA | 42.2 | 39.3 | 34.1 | 26.9 | 22.7 |
| | $S^2$-attn | 45.7 | **41.6** | 36.2 | 30.7 | 22.8 |
| | SelfExtend | 42.0 | 39.4 | 32.4 | 26.8 | 22.4 |
| | NSA | 41.7 | 29.7 | 23.2 | 19.7 | 17.0 |
| | Base | 42.3 | 39.4 | 34.2 | 23.2 | 12.8 |

*Table 10.* Task-wise performance comparison of *L2A* and baseline methods across long-context benchmarks on Qwen3 8B model. *L2A* remains close to CLP performance while outperforming baselines on most tasks. All scores are reported on a 0–100 scale.

| Task | Model | Context Length | | | | |
|---|---|---|---|---|---|---|
| | | **8k** | **16k** | **32k** | **64k** | **128k** |
| Synthetic recall | CLP | **100.0** | **99.9** | **99.7** | **98.8** | 87.9 |
| | L2A (Ours) | 99.8 | 99.8 | 99.4 | 98.2 | 87.1 |
| | FlexPrefill | 99.4 | 99.5 | 99.0 | 98.4 | **88.6** |
| | DCA | 99.2 | 99.4 | 95.4 | 52.7 | 40.0 |
| | Base | 99.1 | 99.4 | 97.4 | 40.0 | 0.0 |
| Retrieval-Augmented Generation | CLP | **65.4** | **63.6** | **60.5** | **59.1** | **52.2** |
| | L2A (Ours) | 63.2 | 61.2 | 60.5 | 58.0 | 51.0 |
| | FlexPrefill | 63.0 | 60.2 | 58.0 | 56.8 | 46.5 |
| | DCA | 63.5 | 61.2 | 57.3 | 52.8 | 47.9 |
| | Base | 63.5 | 61.5 | 58.0 | 47.9 | 7.0 |
| Many-shot in-context learning | CLP | 70.8 | 74.7 | 79.1 | 82.8 | 85.4 |
| | L2A (Ours) | 64.4 | 67.3 | 74.4 | 80.2 | **85.7** |
| | FlexPrefill | **71.2** | **75.8** | **80.2** | **83.7** | 85.1 |
| | DCA | 65.4 | 68.5 | 70.5 | 72.9 | 76.2 |
| | Base | 65.5 | 68.9 | 70.1 | 68.8 | 9.3 |
| Generation w/ citations | CLP | 32.6 | 26.6 | **17.0** | **12.3** | **5.9** |
| | L2A (Ours) | 30.4 | 27.1 | 15.2 | 5.2 | 2.1 |
| | FlexPrefill | 19.3 | 7.7 | 5.7 | 5.0 | 3.6 |
| | DCA | 33.3 | 26.8 | 16.4 | 3.9 | 1.6 |
| | Base | **34.8** | **27.3** | 15.2 | 6.4 | 0.7 |
| Passage re-ranking | CLP | **50.0** | 36.3 | 30.1 | 20.4 | **11.2** |
| | L2A (Ours) | 49.0 | 36.7 | 25.9 | 15.6 | 7.5 |
| | FlexPrefill | 42.3 | 32.4 | 24.3 | 12.1 | 7.0 |
| | DCA | 47.5 | 37.4 | 27.6 | 13.9 | 9.3 |
| | Base | 46.3 | **37.8** | **32.8** | **20.7** | 2.1 |
| Multi-round co-reference resolution | CLP | **38.5** | 33.9 | 31.9 | 33.9 | **30.1** |
| | L2A (Ours) | 36.3 | 34.1 | 31.2 | **34.6** | 26.7 |
| | FlexPrefill | 34.0 | 29.8 | 31.9 | 31.6 | 25.5 |
| | DCA | 35.7 | **35.5** | **33.4** | 29.2 | 17.2 |
| | Base | 35.7 | 34.9 | 32.5 | 29.9 | 8.9 |
| Fact chaining & relational reasoning | CLP | 43.4 | 43.6 | 46.6 | 37.8 | 35.6 |
| | L2A (Ours) | 47.2 | **47.6** | **47.6** | 38.6 | **37.8** |
| | FlexPrefill | 44.8 | 39.0 | 39.0 | 34.2 | 31.0 |
| | DCA | **52.8** | 47.0 | 46.0 | **40.8** | 30.0 |
| | Base | **52.8** | 47.2 | 44.0 | 27.6 | 0.4 |
| Average across tasks | CLP | **57.2** | **54.1** | **52.1** | **49.3** | **44.0** |
| | L2A (Ours) | 55.7 | 53.4 | 50.6 | 47.2 | 42.5 |
| | FlexPrefill | 53.4 | 49.2 | 48.3 | 46.0 | 41.0 |
| | DCA | 56.8 | 53.7 | 49.5 | 38.0 | 31.7 |
| | Base | 56.8 | 53.9 | 50.0 | 34.5 | 4.1 |

*Table 11.* Zero-shot performance on standard short-context benchmarks for Qwen 2.5 models at different scales. In these set of results, we compare CLP for Attention and LayerNorm parameters only vs. training all parameters. At both model scales, training Attn + LayerNorm parameters only is the better choice.

| Scale | Method | BoolQ acc ↑ | CommSenseQA acc ↑ | PIQA acc_n ↑ | Winogrande acc ↑ | ARC-E acc_n ↑ | ARC-C acc_n ↑ | MMLU acc ↑ | SWDE contains ↑ | Avg |
|---|---|---|---|---|---|---|---|---|---|---|
| 1.5B | CLP (Attn + Norm) | **73.36** | **74.45** | 75.95 | 64.72 | **88.72** | **73.81** | **60.46** | 86.50 | **74.74** |
| | CLP (Full Model) | 63.67 | 72.65 | **76.01** | **64.80** | 88.05 | 72.53 | 59.29 | **87.58** | 73.07 |
| 7B | CLP (Attn + Norm) | **82.57** | **84.52** | 79.43 | **76.01** | **96.09** | **88.65** | **73.39** | **91.00** | **83.95** |
| | CLP (Full Model) | 79.60 | 82.88 | **79.76** | 74.03 | 95.54 | 87.88 | 72.76 | 89.56 | 82.75 |

