# OpenReview forum: "Learning When to Attend: Conditional Memory Access for Long-Context LLMs"
_ICML.cc/2026/Conference — ICML 2026 regular_

### Official Review · Reviewer_1iwi · 2026-03-10

**Soundness:** 2
**Presentation:** 3
**Significance:** 3
**Originality:** 2
**Overall Recommendation:** 4
**Confidence:** 3

**Summary:**

This paper proposes L2A, which trains a router to dynamically determine whether each token should perform global attention and reduces the overhead of long-context inference. In addition, the paper improves system efficiency through customized GPU kernels. Experimental results show that L2A maintains high inference accuracy while improving training throughput.

**Compliance With Llm Reviewing Policy:**

Affirmed.

**Final Justification:**

The authors' rebuttal addressed my main concerns. Please add more models in evaluation and improve the presentation of this paper.

**Key Questions For Authors:**

1.	Please justify the generality of the proposed solution.
2.	Please discuss whether the proposed method can work will existing method.
3.	I would like to better understand the overhead of dynamically constructing Qc and Qidx based on dt. Could the authors provide a latency breakdown of the L2A kernel execution?
4.	Please report the training costs of the different training-based methods, such as training time and other relevant overheads.

**Limitations:**

Yes

**Strengths And Weaknesses:**

Strengths:
+ The paper is well-structured and easy to follow.
+ The proposed idea of using a dedicated layer to determine at each token whether to apply Global Attention is reasonable.
+ The authors have designed customized GPU kernels that effectively translate sparsity into practical system efficiency gains.
+ Compared with baseline methods, the proposed approach preserves inference accuracy while attaining remarkable inference acceleration.

Weaknesses:
- The generality of the proposed method is weak. It is questionable whether the examples used to claim the necessity of token-wise adaptivity are representative enough in practice (formal dataset analysis may help). It is also questionable whether the proposed method can performance well for more advanced models.
- More details of the inference performance of L2A should be disclosed. The additional layer introduced by L2A may inflate the inference time, which may be non-negligible for short-context queries. The kernel execution overhead breakdown should be reported.
- Regarding the training overhead, the training costs of the training-based methods should be reported in more detail. Meanwhile, it is questionable whether the standard next-token prediction objective is sufficient for training reasoning models with L2A.
- Can the proposed method (compute exact Attention over the full context but only for a sparse subset of query tokens) be combined with existing sparse-attention method (compute Attention for every query over a sparsified set of past keys). Given the popularity of the latter solution family, the authors shall series discuss the possibility of such combination.
- Some of the solution design details are not solid. When leveraging sparsity for post-training layer KV cache pruning, the authors base their solution on the observation on certain models (e.g., nearly 50% layers in the Qwen 2.5 1.5B L2A model rely exclusively on Local Attention). Can that threshold/method work for general models? Why? Besides, new hyperparameter are introduced in L2A training. While the authors discussed them in Appendix, there is still a lack of clear guidance on how the hyperparameter should be set.
- The evaluation part could be enhanced. Some of the baseline methods are not clearly elaborated. Only mentioning their name is insufficient. The models used are relatively small and narrow (all from the same family). There is a lack of clear summarization presentation of the L2A performance benefit as a trade-off method (the accuracy and efficiency performance are independently elaborated in separated subsections). There is a lack of in-depth sensitivity analysis on when will A2L perform better (e.g., on what type of dataset compositions). The authors shall formally elaborate the negative impact of A2L with a subsection.

---

> ### Author Rebuttal · Authors · 2026-03-31
>
> ## **The generality of the proposed method is weak**
> We respectfully disagree with the reviewer. We demonstrate the generality of our method by evaluating it across long-context benchmarks that reflect real-world usage, including retrieval-augmented generation, many-shot in-context learning, and fact chaining and relational reasoning over long documents. These evaluations are conducted across multiple model scales (1.5B and 7B) and model families (Qwen2.5 and Qwen3). Details of all tasks are provided in Appendix F.2. Across all tasks and model scales, L2A remains competitive with standard Attention-based Transformer baselines in terms of performance, while achieving nearly 2× faster training. Furthermore, in Section 4.4, we analyze sparsity patterns across different task types and show that they align with intuitive expectations, that sparsity adapts to the complexity of the retrieval requirements. We believe that this sufficiently elucidates the generality of our proposed method in enabling token-wise adaptation of Attention across different tasks.
>
> ## **Combination with Existing KV-sparse Approaches**
> Yes, L2A can be combined with existing KV-sparse attention methods. Prior approaches (e.g., NSA, $S^2$-Attn) reduce computation by restricting which keys/values are attended to (i.e., what to attend). In contrast, L2A introduces sparsity along the query dimension by learning when to invoke global attention.
> These two forms of sparsity are orthogonal and can be naturally combined. L2A first selects a subset of query tokens requiring long-range context, after which a KV-sparse method can restrict attention to relevant keys/values for those queries.
> This results in a two-stage mechanism, reducing both the number of active queries and the attention span per query. While our work focuses on query sparsity, exploring such hybrid designs is a promising direction for future work.
>
> ## **Runtime Breakdown of L2A Kernel at Short-Context Lengths**
> **Table 4.1 Forward pass runtime breakdown of L2A**
> | Seq Len | FlashAttn (ms) | Sparsity | L2A Total (ms) | Preprocessing (%) | Speedup |
> |--------|----------------|----------|----------------|------------------|---------|
> | 4096   | 0.41           | 0.20     | 0.59           | 45.40            | 0.70×   |
> |        |                | 0.50     | 0.49           | 56.40            | 0.83×   |
> |        |                | 0.80     | 0.40           | 63.10            | 1.02×   |
> |        |                | 0.90     | 0.37           | 65.80            | 1.10×   |
> |        |                | 0.95     | 0.36           | 66.31            | 1.13×   |
> |
> | 8192   | 1.38           | 0.20     | 1.52           | 32.23            | 0.90×   |
> |        |                | 0.50     | 1.14           | 36.07            | 1.21×   |
> |        |                | 0.80     | 0.75           | 50.16            | 1.84×   |
> |        |                | 0.90     | 0.63           | 58.72            | 2.19×   |
> |        |                | 0.95     | 0.59           | 65.10            | 2.33×   |
> |
>
>  Note that the L2A kernel-level speedups reported in Figures 2 and 8 (in our submission) account for both the preprocessing stage and the sparse attention kernel. We add breakdown of L2A's runtime in Table 4.1, explicitly highlighting the preprocessing (i.e., dynamically constructing the compacted queries $Q_c$ and index tensors $Q_{\text{idx}}$) overhead for short-context lengths.
>  The reviewer is right that at shorter context lengths (e.g., 4K), a higher fraction of runtime (45--66\%) dominates the total runtime of our L2A forward pass. However, at higher sparsity levels, this pre-processing overhead is compensated by the sparser Attention compute. In practice, our L2A models on short-context tasks achieve a very high sparsity level of 90+\% (as measured on LM-Harness tasks reported in the paper). At this level of sparsity, even at 4K context length, our kernels are faster than standard Attention.  Finally, we emphasize that the core contribution of our work is enabling long-context LLMs with significantly reduced training cost, which remains a key bottleneck.
>
> ## **Training Cost of Training-Based Methods**
> We have reported the training cost of L2A vs. standard Attention-based Transformers (CLP) in Table 5 (Appendix). We report training throughput for other training-based methods on Qwen 2.5 7B:
> 1. S$^2$-Attn achieves a throughput of $\sim$1725 tokens/GPU/s, compared to $\sim$1384 tokens/GPU/s for L2A. However, S$^2$-Attn has a $\sim$10 point gap with CLP (our upper bound).
> 2. NSA achieves a throughput of $\sim$565 tokens/GPU/s. This is due to inefficiencies in the publicly available implementation (discussed in Section E). The official implementation has not been released. Moreover, NSA exhibits a substantial performance gap of $\sim$20.5 points compared to CLP.
>
> L2A is the only method (amongst our baselines) that is Pareto-optimal, matching the performance of CLP while being nearly 2x faster to train at 128K.

---

> > ### Author Rebuttal · Reviewer_1iwi · 2026-04-04
> >
> > Thanks for your rebuttal, which has addressed most of my key concerns. I will raise my score to 4. Please try include more diverse models (other than Qwen and of larger sizes > 30B) in the future version of this paper.

---

### Official Review · Reviewer_Yh6z · 2026-03-11

**Soundness:** 3
**Presentation:** 3
**Significance:** 3
**Originality:** 3
**Overall Recommendation:** 5
**Confidence:** 4

**Summary:**

This paper describes an attention mechanism that primarily relies on local context while selectively invoking global attention when necessary. Local attention is implemented using sliding‑window attention, and global attention uses full attention. A routing module, conditioned on the representations produced by local attention, determines when global attention should be activated. Custom Triton kernels are used to ensure efficient computation.During training, the method applies a regularizer to discourage excessive use of global attention and incorporates strategies to prevent router collapse where the model overuses local attention because gradients are not propagated through global attention paths. For efficient inference, layers that infrequently require global attention prune their KV caches, achieving up to 50% memory savings.  Experiments are conducted on two Qwen models - Qwen2.5‑7B and Qwen3‑8B. These base models are further pre‑trained on publicly available datasets with the new attention mechanism enabled (training only token-mixing layers while keeping FFN layers frozen). The approach is evaluated across a range of tasks, including information recall, retrieval‑augmented generation (RAG), and in‑context learning, and is compared against recent attention methods such as SelfExtend, S², and NSA. The method outperforms baselines on most tasks with no deterioration on short-context tasks.

**Compliance With Llm Reviewing Policy:**

Affirmed.

**Final Justification:**

It would be helpful to include more about the training data but otherwise I would like to retain my accept score. The authors promise to release all code and training data scripts for reproducibility.

**Key Questions For Authors:**

- See weakness
- Will the code, data, kernels and models be released?
- Any reason why both models were Qwen?

**Limitations:**

yes

**Strengths And Weaknesses:**

**Strengths**

- Simple idea
- Custom kernels, study of inference characteristics included
- Extensive experiments with natural research questions explored
- Well written paper

**Weakness**
- Some implementation details are missing such as the exact dataset used for training the attention module, performance breakdown on different long-context benchmarks

---

> ### Author Rebuttal · Authors · 2026-03-31
>
> We thank the reviewer for their positive feedback on the efficacy and simplicity of our approach. Next, we address each of the reviewers' comments.
>
>
> ## **Performance breakdown on different long-context benchmarks missing**
> We have provided a detailed performance breakdown on different long-context benchmarks in Tables 2, 3, and 4 (in the Appendix), with per-task average over various context lengths reported in Figure 3. Descriptions of the individual tasks are included in Section F.2.
>
> ## **Why only Qwen models?**
> We used Qwen since at the time of our work, they were the most widely used open-source models at all scales.
>
> ## **Open Sourcing L2A**
> Yes, we plan to release the full codebase, including training scripts and custom kernels. The release will include all components necessary to reproduce our results.

---

> > ### Author Rebuttal · Reviewer_Yh6z · 2026-04-05
> >
> > Thank you for the additional details. It would be helpful to include more about the training data - I would like to retain my accept score.

---

> > > ### Author Response · Authors · 2026-04-08
> > >
> > > Thank you for the question. We provide additional details on our training data composition below.
> > >
> > > Our training data consists of 25B tokens curated from instruction-following data, math, code and heavily filtered web documents sourced from publicly available datasets. The mixture is biased toward longer sequences, with approximately 75% of samples being long documents (≥ 50K tokens) and the remaining 25% being shorter documents.

---

### Official Review · Reviewer_FcWf · 2026-03-11

**Soundness:** 3
**Presentation:** 3
**Significance:** 3
**Originality:** 3
**Overall Recommendation:** 5
**Confidence:** 2

**Summary:**

The author proposes L2A, a sequence modeling layer that enables token-wise long-term conditional memory access by deciding when to invoke Global Attention in this paper. Besides, the authors evaluate the proposed L2A on multiple different models and settings, e.g., Qwen2.5/Qwen 3 models. Rich experimtns resutls denotes propsoed L2A can achieve promising performance gains and ∼2× improvements in training throughput, and improve performance on long-horizon reasoning and retrieval of current LLMs.

**Compliance With Llm Reviewing Policy:**

Affirmed.

**Final Justification:**

In this rebuttal, my concerns have been fully addressed. Therefore, I will raise my original score.

**Key Questions For Authors:**

Please refer to the Weaknesses above. Thanks.

**Limitations:**

Yes! The authors provide some limitations in the discussion section.

**Strengths And Weaknesses:**

一、Strengths

In this paper, to improve performance on long-horizon reasoning and retrieval of current LLMs, the author proposes L2A, a sequence modeling layer that enables token-wise long-term conditional memory access by deciding when to invoke Global Attention. The authors evaluate the proposed L2A on multiple different models and settings, e.g., Qwen2.5/Qwen 3 models. Besides, rich experimtns resutls denotes propsoed methods can achieve promising performance gains and ∼2× improvements in training throughput.  In total, I think this paper is well-designed, with a clear presentation and motivation, and has a significant influence on future research.

二、Weaknesses

(1) In lines 165-168, the author mentions that L2A increases the number of parameters by approximately 10% and reduces the KV cache during inference. Does this performance improvement stem solely from “increasing the number of parameters and continuous training”? The authors should supplement with a baseline experiment. If the Base LLM were similarly augmented with 10% more parameters (e.g., via a larger FFN or additional layers) and underwent the same scale of continued pre-training, would L2A's advantage remain significant? Current experiments mainly compare training-free methods, which is not entirely fair to L2A, which requires additional training and increased parameters.

(2) In section 3.4, KV cache savings heavily depend on post-processing pruning. I think the author requires verification of pruning stability for ultra-long sequences.

(3) In Fig.6, certain tasks (such as ICL) exhibit significantly lower sparsity than Recall tasks. If L2A is not sparse in complex tasks (ICL), does its claimed “efficiency advantage” still hold up when confronted with truly challenging tasks?

---

> ### Author Rebuttal · Authors · 2026-03-31
>
> We thank the reviewer for their time and feedback. We are glad that the reviewer found our work well-designed, our presentation clear, and our contributions to have significant potential for influencing future research. Next, we address each of the reviewers' comments.
>
> ## **Clarification on Training Baselines**
>
> Our original submission already includes comparisons with two training-based methods (NSA and $S^2$-Attn). As shown in Figures 2, 3, and Tables 2, 3, and 4, L2A consistently outperforms both baselines.
>
> ## **Does L2A's performance improvements stem solely from “increasing the number of parameters and continuous training”?**
>
> We now have experiments with the QKVO projection parameters shared between the Local and Global Attention modules. This removes the additional ~10\% overhead in the parameter count. We call this variant L2A-shared and find it to be competitive with L2A in terms of performance on long context tasks. Detailed results are reported in Table 2.1, where we compare the following:
> - CLP: Transformer with standard attention trained at long context.
> - L2A: Our proposed method.
> - L2A-shared: L2A with shared QKVO projection layers.
> All models use Qwen2.5 1.5B as the base LLM and are trained under the same setup described in Appendix F.1.
>
> **Table 2.1 Average performance across tasks at different context lengths (%)**
>
> | Model       | 8k   | 16k  | 32k  | 64k  | 128k |
> |------------|------|------|------|------|------|
> | CLP        | 45.7 | **41.4** | **37.6** | **35.0** | **29.8** |
> | L2A        | **45.9** | 40.7 | 36.5 | 34.2 | 29.6 |
> | L2A-shared | 43.8 | 40.2 | 36.0 | 32.7 | 27.6 |
>
> Note, a difference of 1–2 points in performance is within the variance reported for this benchmark (see https://github.com/princeton-nlp/HELMET/issues/7#issuecomment-2435378761).
>
> ## **Verification of KV cache saving technique in Section 3.4**
> We have carried out this experiment in Section 4.3 in our original submission, with results reported in Figure 18 for the Qwen2.5 1.5B model. Our pruning technique is based on held-out data from the training set and is not adapted to any specific downstream task. Specifically, we compute the average sparsity for all the layers on this held-out-data and drop layers that are $\geq$95\% sparse. This results in nearly 50\% of layers being dropped for the Qwen 2.5 1.5B L2A model with minimal impact on performance up to 128k context.
> We further extend this analysis to the Qwen2.5 7B model and observe similar trends. Applying the same $\geq$95\% sparsity threshold, we are able to prune $\sim$40\% of the layers while maintaining comparable performance, along with substantial KV cache savings. We report these results in Table 2.2 using Qwen2.5-7B as the base LLM. CLP and L2A are defined as in Table 2.1, and L2A [Pruned] denotes the L2A model after applying the above pruning procedure.
>
> **Table 2.2 Average performance across tasks at different context lengths (%) (Qwen 2.5-7B as the base LLM)**
>
> | Model        | 8k   | 16k  | 32k  | 64k  | 128k |
> |-------------|------|------|------|------|------|
> | CLP-7B      | **56.0** | **52.1** | **49.0** | 45.4 | 40.7 |
> | L2A         | 53.8 | 51.4 | 48.3 | **45.6** | **41.6** |
> | L2A [Pruned]| 52.9 | 50.2 | 46.7 | 45.2 | 40.6 |
>
> ## **L2A has low sparsity for the ICL task. Does its claimed “efficiency advantage” still hold up when confronted with truly challenging tasks?**
> The observed difference in sparsity across tasks is, in fact, an intended behavior of L2A.
>
> L2A is designed with the insight that not all tokens require looking back at the entire past (via Attention) for processing, and thus the model can adaptively control this cost of calling Attention depending on the "complexity" of the task and context. For example, for recall tasks like Needle-in-a-Haystack, long-term memory access is typically required only at a few specific token positions, whereas
> for tasks like in-context learning, Global Attention may be invoked
> frequently to compare and reuse provided examples. This behavior is discussed in detail in Section 4.4. Moreover, even for ICL, the average sparsity at 128K is about 60\% (see Figure 6 in the main paper), which still results in 1.6x speedup over the standard Attention baseline.

---

> > ### Author Rebuttal · Reviewer_FcWf · 2026-04-02
> >
> > Thank you for your detailed response! My concerns have been fully addressed. Therefore, I will raise my original score.

---

### Official Review · Reviewer_juEZ · 2026-03-13

**Soundness:** 3
**Presentation:** 3
**Significance:** 3
**Originality:** 3
**Overall Recommendation:** 4
**Confidence:** 4

**Summary:**

This paper proposes a novel architecture, called L2A, in which each layer consists of a global attention and a local attention, where the global attention is only computed for a subset of tokens. This model is considerably faster than ordinary attention in long-context scenarios, and it outperforms existing training-free sparse attention, training-free context extension, and training-based sparse attention methods in terms of performance-efficiency tradeoff. This method has great practical value for long-context processing using LLMs.

**Compliance With Llm Reviewing Policy:**

Affirmed.

**Final Justification:**

This paper proposes a novel architecture as an efficient alternative to vanilla attention. Empirical results show that the model outperforms vanilla attention mechanisms in terms of performance-efficiency tradeoff. The authors have resolved my concerns during the rebuttal period, and I appreciate the empirical value of the paper. I will maintain my positive score of the paper.

**Key Questions For Authors:**

1. Currently, the proposed method only uses 25B or fewer tokens to adapt a pre-trained Transformer model into L2A. I want to know if this model can be trained from scratch and still enjoy the performance/efficiency advantages. Adding such clean from-scratch pre-training experiments would greatly strengthen the paper's value.
2. The proposed method increases the model's parameter count (there are sets of attention-related parameters in each layer instead of one). How is the performance of L2A if we instead shared the QKVO projections for the global and local attention modules?

**Limitations:**

yes

**Strengths And Weaknesses:**

Strengths:

- The proposed method is simple yet effective and efficient, outperform many state-of-the-art methods.
- The paper is well-written and generally easy-to-follow.
- The evaluation experiments are comprehensive, and the models involved have sufficient scale to ensure that the results are applicable to real-world scenarios.
- The authors implemented a custom kernel to utilize the strengths of this novel architecture and empirically demonstrated superior inference and training throughput.

Weaknesses:

- While the authors have already evaluated some popular training-based sparse attention methods (NSA and S$^2$-Attn), these are already one year old and I think the evaluation results would be more convincing if they compared against some more recent training-based sparse attention methods, such as InfLLM-v2, DSA (DeepSeek Sparse Attention), etc.
- The proposed model is only evaluated in the setting where a pre-trained Transformer model is converted into a sparse attention model through post-training with limited data, but a from-scratch training may be cleaner and more valuable for many practitioners.

---

> ### Author Rebuttal · Authors · 2026-03-31
>
> We thank the reviewer for their time and feedback. We are glad that the reviewer found our work to have great practical value to long context LLMs, the paper well-written and easy to follow. Next, we address the reviewers' comments.
>
> ## **Pretraining from Scratch**
> We focus on developing long-context methods for off-the-shelf Transformer models, motivated by the high cost of pretraining, which often requires trillions of tokens to obtain a strong model [1].
> Nevertheless, we acknowledge the value of training from scratch and train both a standard Transformer and L2A using the Qwen2.5 1.5B. Models are trained on 100B tokens from DCLM [2] at 4K context, followed by continued pretraining at 128K, and evaluated on LM-Harness and long-context benchmarks using the same setup as in the main paper.
>
> **Table 1.1 Short-context performance**
> | Task           | Transformer | L2A   |
> |----------------|------------|-------|
> | BoolQ          | 60.86      | 55.66 |
> | CommonSenseQA  | 19.25      | 19.16 |
> | MMLU           | 25.80      | 25.14 |
> | PIQA           | 71.76      | 71.33 |
> | SWDE           | 65.98      | 65.89 |
> | Winogrande     | 58.56      | 59.04 |
> | ARC-C          | 24.83      | 25.17 |
> | ARC-E          | 25.51      | 24.41 |
> | **Average**    | **44.06**  | **43.23** |
>
> **Table 1.2 Long-context performance**
> | Model       | 8k   | 16k  | 32k  | 64k  | 128k |
> |------------|------|------|------|------|------|
> | Transformer| 15.0 | 14.0 | 12.4 | 11.0 | 10.1 |
> | L2A        | **17.0** | **15.7** | **14.4**  | **13.0**  | **11.8**  |
>
> L2A performs within 1 point of the Transformer on LM-Harness (Table 1.1) while remaining competitive on long-context benchmarks (Table 1.2). It achieves an average sparsity of ~60\% across tasks, resulting in ~1.25x training speedup and ~1.6x improvement in average time-to-first-token during inference at 128K context.
> Overall, these results highlight L2A’s efficiency gains even when trained from scratch, despite no additional parameter tuning.
>
> 1. Qwen Team, "Qwen 2.5 Technical Report", arXiv:2412.15115 (2025)
> 2. Li et al., "DataComp-LM: In search of the next generation of training sets for language models", arXiv:2406.11794 (2025).
>
> ## **L2A with Shared QKVO Projections**
> We evaluate a variant of our method with shared QKVO projection layers, using Qwen2.5 1.5B as the base LLM.
> We compare CLP (Transformer with standard attention trained at long context), L2A (our method), and L2A-shared (L2A with shared QKVO projection layers).
>
> **Table 1.3. Average long-context performance**
> | Model      | 8k   | 16k  | 32k  | 64k  | 128k |
> |------------|------|------|------|------|------|
> | CLP        | 45.7 | **41.4** | **37.6** | **35.0** | **29.8** |
> | L2A        | **45.9** | 40.7 | 36.5 | 34.2 | 29.6 |
> | L2A-shared | 43.8 | 40.2 | 36.0 | 32.7 | 27.6 |
>
> The small performance gap of 1-2 points between L2A and L2A-shared falls within the variance reported for this benchmark (https://github.com/princeton-nlp/HELMET/issues/7#issuecomment-2435378761). Both L2A variants remain close to CLP, indicating that parameter sharing does not degrade performance. We analyze sparsity in Table 1.4 (i.e, fraction of tokens invoking Global Attention)
>
> **Table 1.4. Average sparsity (%)**
> | Model       | 8k   | 16k  | 32k  | 64k  | 128k |
> |------------|------|------|------|------|------|
> | L2A        | 94.5 | 91.3 | 86.9 | 82.2 | 78.1 |
> | L2A-shared | 88.0 | 82.4 | 76.2 | 70.6 | 66.2 |
>
> While L2A-shared remains competitive with L2A in terms of performance, it achieves up to 10\% lower sparsity. We attribute this to parameter sharing, where the same projections must support both Local and Global Attention, limiting their ability to specialize for selective routing.
>
> ## **Comparison with recent training-based baselines**
> We are unable to provide a fair end-to-end comparison with these methods due to limitations in the publicly available implementations.
> For InfLLM-v2, although training code is available, the kernel requires the group size in Group Query Attention to be a multiple of 16 (https://github.com/OpenBMB/infllmv2_cuda_impl/issues/3), which is incompatible with the Qwen configurations used in our experiments. For DSA, although forward-pass kernels are available (https://github.com/deepseek-ai/FlashMLA/pull/98
> , https://github.com/deepseek-ai/DeepGEMM/pull/200
> ), training code and backward-pass implementations are not publicly released.
>
> Due to these limitations, we compare kernel-level efficiency as our L2A models match the performance of ordinary Attention (upper bound for performance) while achieving significantly faster training. In particular, our kernels achieve up to a 5.5× speedup at 128k context length, with average sparsity of $80\%$ across tasks (Figure 8), whereas InfLLM-v2 reports up to a 4× speedup at the same context length (Figure 6 in [3]) compared to ordinary Attention.
>
> 3. Zhao, et al. "Infllm-v2: Dense-sparse switchable attention for seamless short-to-long adaptation." arXiv:2509.24663 (2025).

---

> > ### Author Rebuttal · Reviewer_juEZ · 2026-04-03
> >
> > I appreciate the additional experiments by the authors, and I think the paper has great empirical value. I will maintain my positive score.

---

### Decision · Program_Chairs · 2026-04-30

**Decision:**

Accept (regular)

**Comment:**

This paper proposes L2A, a drop-in attention layer that learns token-wise conditional memory access, enabling efficient long-context extension from 32K to 128K tokens. All four reviewers found the paper well-written with a simple yet effective idea, comprehensive experiments across multiple model scales (Qwen 2.5 and Qwen 3), and practical contributions including custom Triton kernels achieving ~2× training speedup and ~50% KV cache savings.

Reviewers raised concerns regarding generality across model families, from-scratch training, the overhead of added parameters, and comparisons with recent sparse attention methods. The authors provided thorough rebuttals — including from-scratch pre-training results, shared-projection ablations, kernel runtime breakdowns, and training cost comparisons — which were acknowledged by all reviewers (three fully resolved, one partially resolved while retaining the accept score). Final scores converged to 4–5, with no reviewer below weak accept, and thus I recommend acceptance.